# Somatic hypermutation patterns are shaped by both motif position and sequence grammar

Bianca Bartl[1,8], Ursula E Schoeberl [ID][1,2,3,8], Renan Valieris [ID][4,8], Johanna Fitz[1], Konstantin Roeder [ID][5], Kutti R Vinothkumar [ID][6], Benjamin Gundinger [ID][1], Israel Tojal Da Silva [ID][4] & Rushad Pavri [ID][1,7 ✉]

## Abstract

Somatic hypermutation (SHM) in variable regions of immunoglobulin genes by activation-induced deaminase (AID) is essential for the maturation of protective antibodies against pathogen and vaccine antigens. AID preferentially mutates cytosines within WRCH motifs (wherein W = A/T, R = A/G, and H = A/C/T) in single-stranded DNA, yet these motifs show large but reproducible variation in mutation frequency, suggesting a crucial role for sequences flanking the WRCH motifs (i.e., a sequence grammar) in determining mutational outcomes. However, the nature of this sequence grammar is poorly understood. Here, we demonstrate that identical sequence contexts can exert significantly varying effects on the mutagenesis of different WRCH motifs. Molecular dynamics simulations reveal that both the sequence context and the specific WRCH motif modulate AID activity by altering the mode and strength of AID's interactions with single-stranded DNA. Repositioning a motif and its context within the variable region significantly alters its mutability. Therefore, the mutability of AID target cytosines is determined by a motif-specific sequence grammar that determines, in part, how activation-induced deaminase binds single-stranded DNA, as well as the motif position.

**Keywords** Somatic Hypermutation; Activation Induced Deaminase; DNA Sequence Grammar; Ramos Cells; Molecular Dynamics Simulation
**Subject Categories** DNA Replication, Recombination & Repair; Immunology

## Introduction

Somatic hypermutation (SHM) is the molecular basis for all long-term serum immunity via its central role in diversifying antibodies in response to pathogens and vaccines. Mutations are generated in the variable (V) regions of the immunoglobulin heavy (IGH) and light chain (IGK,IGL) genes by activation-induced cytidine deaminase (AID) (Muramatsu et al, 2000; Revy et al, 2000), which acts co-transcriptionally on single-stranded DNA (ssDNA) (Bransteitter et al, 2003; Petersen-Mahrt et al, 2002; Pham et al, 2003; Ramiro et al, 2003). SHM occurs in antigen-activated B cells within germinal centers of secondary lymphoid tissues and results in the alteration of antigen-antibody binding affinities, with B cells expressing antibodies with higher affinity being preferentially selected for clonal expansion (Victora and Nussenzweig, 2022).

One of the major questions in SHM biology pertains to the discrete mutation patterns of V regions. AID preferentially deaminates cytosines to uracils within WRCH motifs (where W = A or T, R = A or G and H = A, C or T) (Methot and Di Noia, 2017). However, mutation profiles are marked by the differential mutation of WRCH motifs and a tendency for higher mutability of the antigen-binding complementarity-determining regions (CDRs) relative to the intervening, structural, framework regions (FWRs). This has led to the suggestion that sequence-intrinsic mechanisms guide AID targeting and determine motif mutability (Hwang et al, 2017; Spisak et al, 2020; Wei et al, 2015; Yeap et al, 2015; Zhou and Kleinstein, 2020). Indeed, experiments in mice engineered to express productive and non-productive versions of the same V region found that mutation profiles were indistinguishable between the productive and non-productive alleles, implying that mutation profiles were derived from sequence-intrinsic mechanisms that were independent of affinity selection within germinal centers (Hwang et al, 2017; Yeap et al, 2015). Of note, various co-transcriptional activities such as pausing (Pavri et al, 2010; Rajagopal et al, 2009; Wang et al, 2009a), RNA processing (Basu et al, 2011; Pefanis et al, 2014), convergent transcription (Meng et al, 2014), elongation (Begum et al, 2012; Dai et al, 2025; Methot et al, 2018; Willmann et al, 2012; Wu et al, 2025) and active chromatin modifications (Aida et al, 2013; Begum et al, 2012; Bradley et al, 2006; Daniel et al, 2010; Duan et al, 2021; Jeevan-Raj et al, 2011; Kuang et al, 2009; Stanlie et al, 2010; Vaidyanathan and Chaudhuri, 2015; Yu et al, 2021), have been linked to SHM. However, nascent transcriptional profiling of multiple V regions and hundreds of non-immunoglobulin SHM targets did not reveal any correlation between WRCH mutation frequencies and

[1]Research Institute of Molecular Pathology (IMP), Campus-Vienna-Biocenter 1, 1030 Vienna, Austria. [2]Max Perutz Labs, Vienna Biocenter Campus (VBC), Dr.-Bohr-Gasse 9, 1030 Vienna, Austria. [3]University of Vienna, Max Perutz Labs, Department of Biochemistry and Cell Biology, Dr.-Bohr-Gasse 9, 1030 Vienna, Austria. [4]Laboratory of Bioinformatics and Computational Biology, A. C. Camargo Cancer Center, São Paulo, SP, Brazil. [5]Randall Centre for Cell and Molecular Biophysics, King's College London, London SE1 1UL, UK. [6]National Centre for Biological Sciences, Tata Institute of Fundamental Research, GKVK Post, Bengaluru 560065, India. [7]Peter Gorer Department of Immunobiology, School of Immunology & Microbial Sciences, King's College London, London SE1 9RT, UK. [8]These authors contributed equally: Bianca Bartl, Ursula E Schoeberl, Renan Valieris. ✉E-mail: rushad.pavri@kcl.ac.uk

transcriptional strength or any co-transcriptional feature (Schoeberl et al, 2025a). These results support the notion that, following AID recruitment to V regions, mutational outcomes majorly depend on the sequences flanking each WRCH motif. We refer to this sequence neighborhood or locale henceforth as the sequence context, by which we mean the regulatory information encoded in the DNA sequence that defines the rules governing the magnitude of SHM at WRCH motifs.

Recent work has advanced our understanding of how sequence contexts could regulate AID targeting preferences. Using a high-throughput biochemical ssDNA deaminase assay, Py-rich sequence contexts flanking WRC motifs were found to correlate with higher mutability of WRC motifs (Wang et al, 2023). A synthetic pyrimidine (Py)-rich cassette (TACTAC) inserted upstream of a weakly mutated WRCH motif, AGCT, within the murine B1-8 V region, significantly increased SHM at this motif (Wang et al, 2023). Molecular simulations suggested that ssDNA with a synthetic Py-only (poly-T) sequence upstream of an AGCT motif bound strongly to key positively charged surface residues in AID, specifically, the assistant patch, whereas a synthetic purine-only (poly-A) sequence did not (Wang et al, 2023). It was proposed that weaker base stacking of PyPy dimers relative to purine-purine dimers, could make PyPy-rich ssDNA more flexible for AID targeting (Wang et al, 2023).

Another study used an interpretable deep learning approach to report that SHM may be actively suppressed in some WRCH sequence contexts. Specifically, it was found that conserved AGCT motifs near the 5' ends of human V regions suffer significantly reduced mutagenesis relative to other V region AGCT motifs (Tambe et al, 2024). This dampening of AGCT mutagenesis coincided with the presence of an overlapping E2A-binding E-box motif (CAGCTG) and naturally occurring mutations that ablated this motif (CAGCTH) significantly increased mutagenesis of the associated AGCT motif (Tambe et al, 2024).

These studies raise several questions for further investigation. For example, many WRCH mutational hotspots are not naturally embedded in PyPy-rich neighborhoods at IG and non-IG loci. This includes the AGCT motifs abundant in mammalian immunoglobulin switch recombination sequences that are flanked by G-rich DNA. Moreover, the suppressive effects of E-box motifs were independent of the positive contribution of PyPy density towards AGCT mutability, and both features combined did not fully explain AGCT mutability (Tambe et al, 2024). These observations suggest that additional mechanisms are involved in determining the probability and frequency of SHM. Moreover, since the above-mentioned studies focused on AGCT motifs, the effect of sequence contexts on SHM at other WRCH motifs remains to be deciphered.

Here, using the well-studied murine B1-8$^{hi}$ V region as a template and the human SHM-competent Ramos B cell line as the experimental system, we ask two fundamental questions regarding the nature of the sequence grammar underpinning SHM: (1) Is there a universal SHM grammar for all WRCH motifs, or are there motif-specific grammars? (2) Does the position of a motif within the V region affect its mutation rate? Our results suggest that SHM patterns arise from at least two distinct mechanisms: a motif-specific sequence grammar, governed in part by the mode of ssDNA binding to AID, and the position of the motif within the V region.

# Results

## Analysis of human V regions suggests that sequence contexts favoring AID activity differ between WRCH motifs

To investigate the contribution of PyPy dimers in AID targeting in their natural context, we analyzed a published dataset of ~19,000 non-productive and clonally independent human V region sequences obtained from B cell receptor sequencing (Zhou and Kleinstein, 2020). Since non-productive sequences are not subject to antigen-driven affinity selection pressures within germinal centers, they provide an unbiased view of the intrinsic preferences and biases of the SHM machinery. V regions were divided into 15-mers centered at cytosine residues. When considering all cytosine residues, mutations were generally greater in the CDRs (Fig. EV1A, left panel). This trend was also seen for mutations at cytosines within WRCH motifs, although FWR2 and, to a lesser extent, FWR3, also showed relatively high mutation load (Fig. EV1A, right panel).

To determine how each WRCH motif contributes to this positional bias, the WRCH-centered 15-mers were further divided based on WRCH motif identity, resulting in twelve motif groups (Fig. EV1B). The mutational bias towards CDRs was evident in some WRCH motifs (AGCA, AGCT, TACC, and TGCT) but not in others (AACT and TACA), with some motifs being under-represented or absent in CDRs (AGCC, TGCA, and TGCC) (Fig. EV1B). For most motifs, FWR1 was the least targeted sub-region (Fig. EV1B). Thus, the positional differences between CDR and FWR mutagenesis arise from the relatively higher targeting of selected motifs in CDRs.

To investigate the relationship between PyPy dimers and mutation load, we ranked the 15-mers within each group based on the mutation frequency of the central C residue and counted the number of PyPy dimers in the WRCH ± 5 bp segment (Fig. 1A). We observed no discernible relationship between mutation frequency and PyPy content in general, although a negative relationship was evident for AACT and TACC, wherein 15-mers with the most highly mutated central C residues were devoid of PyPy motifs (Fig. 1A).

We then asked whether PyPy richness affected the probability of mutation of the AID-target C within WRCH motifs. If so, then a positive correlation would be expected between PyPy density and the fraction of mutated sequences. We found that the fraction of 15-mers mutated at the central C residue correlated positively with PyPy content for some motifs (like AGCC and, to a lesser extent, AGCT), negatively with others (AACT in particular) or not at all (AGCA, AACA, and others) (Fig. 1B). Of note, the positive correlation for AGCT mutability and PyPy content is consistent with in vitro biochemical analyses (Wang et al, 2023). Thus, PyPy richness may contribute to the mutability of some WRCH motifs, but it is not a predictor of mutability in general.

Importantly, however, when we inspected the actual mutation frequency of the mutated 15-mers analyzed in Fig. 1B, we observed considerable variations in the mutation frequency of the central C residue even between 15-mers having the same PyPy content, with AGCT-centered 15-mers showing particularly large variations (Fig. 1C). The net result was a poor correlation of mutation frequency with PyPy content for nearly all motifs (Fig. 1C).

A

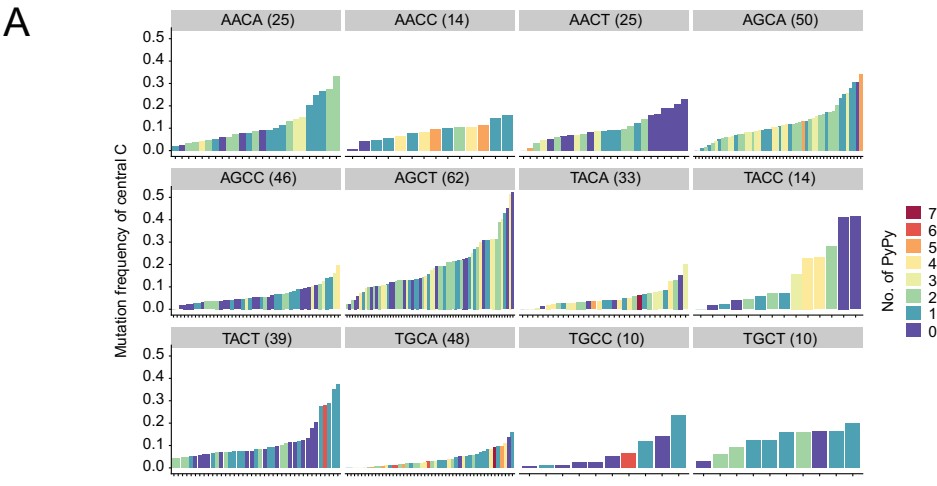

B

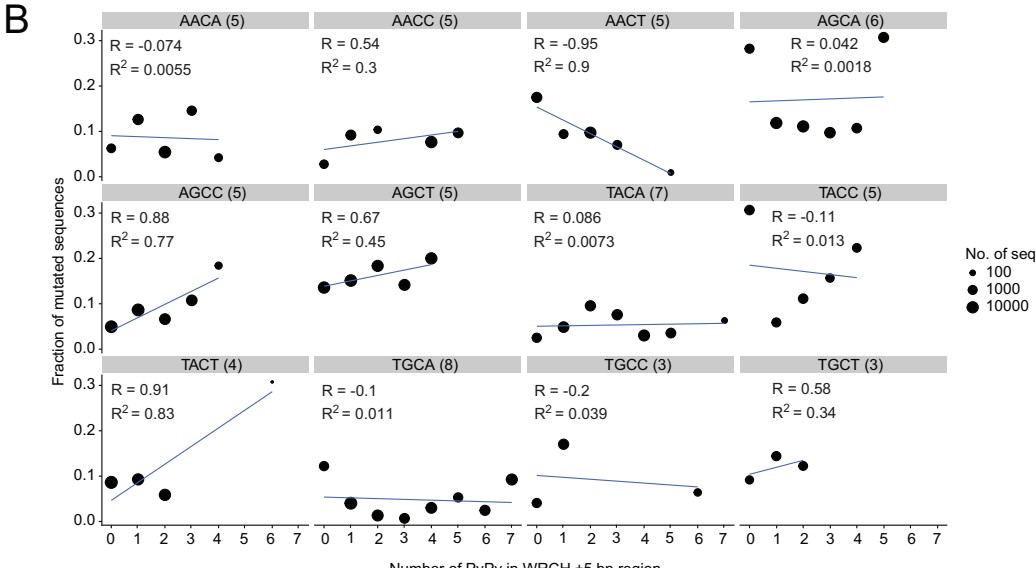

C

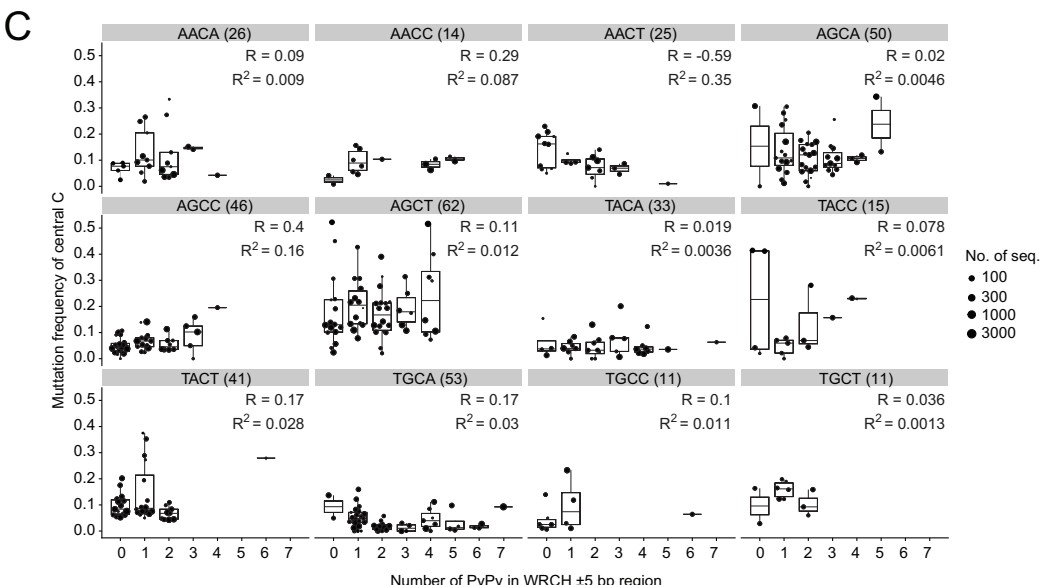

**Figure 1.  Correlation of PyPy content with mutability and mutation frequency of WRCH motifs in human V regions.**

(A) Comparing the mutation frequency of the central C residue of WRCH motifs with PyPy content in the ±5 bp flanking the motif. Each bar represents a unique 15-mer centered at the C within the indicated WRCH. The numbers of unique 15-mers (N) are indicated in brackets. Bars are color-coded to highlight the number of PyPy dimers, as shown in the key on the right. (B) Comparing the fraction of 15-mers with mutations in the central C residue relative to the PyPy content in the ±5 bp sequence flanking the WRCH motif. The numbers of unique 15-mers (N) are indicated in brackets. The size of the dot corresponds to the number of 15-mers. R is the Pearson correlation coefficient. $R^2$ indicates the proportion of the variance in the fraction of mutated sequences arising from the differences in PyPy content. For example, 45% of the variation in AGCT mutability is explained by differences in PyPy content. (C) Box plots showing variation in mutation frequency relative to PyPy content for the four WRCH motifs in A. Each dot represents a unique 15-mer, and the size of the dot indicates the number of such 15-mers in the collection of clonally independent sequences used for these analyses (Zhou and Kleinstein, 2020). The numbers of unique 15-mers (N) are indicated in brackets. Box plots were generated as described in Fig. EV1. The exact P values are shown in Dataset EV1.

We conclude that although PyPy density may increase the probability of mutation at some WRCH motifs in some contexts, it does not appear to influence the actual mutation frequency of AID-target cytosines. More importantly, the results suggest that there exist distinct sequence contexts favoring or disfavoring AID targeting at different WRCH motifs.

## Replacing the major AGCT hotspot in CDR3 of B1-8$^{hi}$ with other WRCH motifs significantly reduces mutability

The above results suggested the possibility that sequence contexts favoring SHM at one type of WRCH motif may not do so at other WRCH motifs. To test this notion empirically, we employed the human IgM$^+$ B cell line, Ramos, an extensively validated model of SHM at IG loci, non-IG AID off-targets and enhancer-driven SHM of reporter substrates (Sale and Neuberger, 1998; Senigl et al, 2019; Tarsalainen et al, 2022; Wu et al, 2022). The mutation profiles in Ramos cells mimic many aspects of physiological SHM, the exception being that mutations occur predominantly at AID-targeted C:G residues with substantially weaker mutagenesis of A:T residues that normally arise from the repair of AID-induced mismatches (Sale and Neuberger, 1998; Schoeberl et al, 2025a). Ramos cells do not require a B cell receptor for survival, which allows for extensive genetic manipulation of V region sequences. These features, and the fact that affinity selection pressures are absent in this in vitro system, results in a mutational landscape that reflects the footprint of AID activity, making Ramos cells well-suited to study how sequence contexts affect AID targeting. For all experiments, SHM was triggered by infection of Ramos cells with lentiviruses expressing a hyperactive mutant of AID, AIDm7.3, which harbors ~3-fold greater catalytic activity relative to wild-type AID (Wang et al, 2009b), followed by mutation analysis by paired-end sequencing (MutPE-seq) (Robbiani et al, 2015; Schoeberl et al, 2025a).

For this study, we chose the murine B1-8$^{hi}$ V region (Shih et al, 2002). We previously showed that, when integrated into the Ramos *IGH* locus in place of the endogenous V region, mutation profiles at C:G residues were comparable to those observed at the endogenous B1-8$^{hi}$ V region in germinal center B cells (Schoeberl et al, 2025a). B1-8$^{hi}$ harbors five palindromic AGCT motifs at the following nucleotide positions: 27G/28C, 54G/55C, 89G/90C, 243G/244C and 311G/312C (Fig. 2A, top row, and indicated by the red bars below the X-axis). These AGCT motifs undergo SHM at different rates (Schoeberl et al, 2025a; Yeap et al, 2015). In particular, the AGCT within CDR3 (AGCT$^{CDR3}$, 311 G/312 C) is the strongest SHM hotspot in B1-8$^{hi}$, whereas the AGCT in FWR3 (AGCT$^{FWR3}$, 243 G/244 C), located 117 bp upstream, mutates with 5–10-fold lower

frequency (Fig. 2A, first row). Of note, since AGCT is palindromic, mutations are observed at both G and C residues, resulting in a double peak in MutPE-seq analyses (Fig. 2A, top row).

We first asked whether the sequence context of AGCT$^{CDR3}$ would confer high rates of SHM at other WRCH motifs, that is, whether the CDR3 locale is privileged for SHM in general. Using our previously described knock-in system in Ramos cells (Schoeberl et al, 2025a), wherein any V region can be inserted in place of the endogenous V region, we generated cell lines expressing variants of B1-8$^{hi}$. Specifically, we replaced AGCT$^{CDR3}$ with five other WRCH motifs: AACT, AGCA, AGCC, TGCA, and TGCT (Fig. 2A, rows 2–6). Nascent V region transcription was comparable between all new cell lines and the Ramos B1-8$^{hi}$ line (Fig. EV2). MutPE-seq showed that, relative to the mutation frequency of AGCT$^{CDR3}$ in wild-type B1-8$^{hi}$ (B1-8$^{hi}$ wt), all the other WRCH substitutions in CDR3 underwent significantly lower SHM (Fig. 2A and quantified in Fig. 2B). Thus, the CDR3 locale seems to favor higher mutagenesis of AGCT compared to other WRCH motifs. These changes were restricted to the altered WRCH locale as exemplified by SHM of the frequently mutated AGCA motif in CDR2 (AGCA$^{CDR2}$, positions 194G and 195C), which is not significantly affected by the changes in CDR3 (Fig. 2B).

Interestingly, TGCA, which, like AGCT, is also palindromic, was the least mutated motif (Fig. 2A row 5 and Fig. 2B), suggesting that being palindromic is not sufficient to confer higher rates of SHM in this locale. Moreover, AGCA and TGCT are WRCH motifs in both orientations, further supporting the notion that the presence of juxtaposed AID-target cytosines are not, by themselves, sufficient to explain the higher mutation rate of AGCT in this CDR3 context. Finally, we note that when AGCT$^{CDR3}$ was replaced by AGCA, this AGCA mutated less frequently than AGCA in CDR2 (Fig. 2A, row 3), which suggests that the CDR2 locale may provide a more favorable context for AGCA mutagenesis than the CDR3 locale in this V region.

These findings suggest the presence of a motif-specific sequence grammar, rather than a motif-agnostic grammar, wherein optimal AID activity is a function of both sequence context and motif identity.

## Further evidence for a motif-specific sequence grammar: sequence contexts supporting AGCT mutagenesis are suppressive for AACT mutagenesis

As seen in our analysis of human V regions in Fig. 1B, mutagenesis of TGCT and AGCT correlated positively with neighboring PyPy content, suggesting that sequence contexts regulating mutagenesis at these two motifs may be compatible. In contrast, AACT mutagenesis was negatively correlated with local PyPy richness (Fig. 1B), suggesting that deamination at AACT and AGCT may be

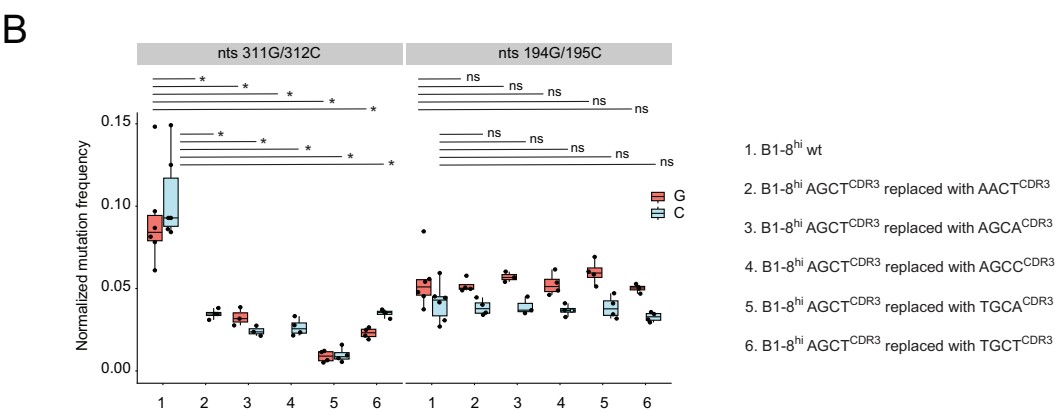

A

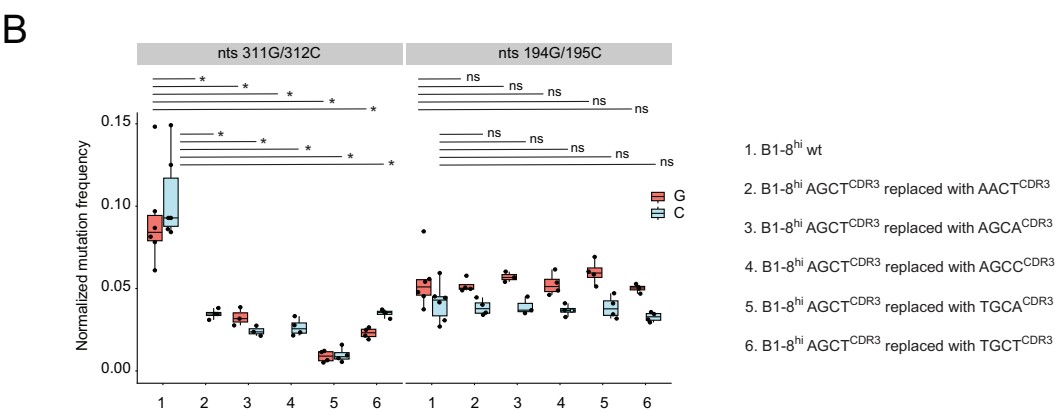

B

◄ **Figure 2. Replacing AGCT^CDR3 in B1-8^hi with other WRCH motifs leads to reduced SHM.**

(A) Mutational profiles of the B1-8^hi wt V region (top row) and a series of clones wherein AGCT^CDR3 was changed to AACT, AGCA, AGCC, TGCA, and TGCT (rows 2–6). Mutations are color-coded as indicated in the key on the right of each plot. The locations of all WRCH motifs are indicated as black bars below the X-axis, with AGCT hotspots shown as red bars. The FWRs and CDRs are indicated, and the CDRs are shaded. The numbers next to the mutated AGCT in CDR3 (311G/312C) and AGCA in CDR2 (194G/195C) denote the positions of the G and C residues within these motifs. Note that AGCT, AGCA, TGCA, and TGCT are WRCH motifs in both orientations and hence show mutations at both G and C residues within the motifs, whereas AACT and AGCC are unidirectional WRCH motifs. (B) Box plots quantifying mutation frequencies at the indicated residues: 311G/312C within AGCT^CDR3 (left panel) and 194G/195C within AGCA^CDR2 (right panel) from six B1-8^hi wt samples ($N = 6$) and four samples from all other clones ($N = 4$). Red and blue boxes represent G and C mutations, respectively. The asterisks indicate $P < 0.05$ using the unpaired Student's t-test. ns, not significant. Box plots were generated as described in Fig. EV1. The exact $P$ values are shown in Dataset EV1.

regulated by different and incompatible sequence contexts. To test this notion, we generated Ramos B1-8^hi lines wherein all five AGCT motifs in wild-type B1-8^hi were changed to TGCT or AACT. No significant changes in V region nascent transcription were observed relative to the Ramos B1-8^hi line (Fig. EV2).

In line with the results in Fig. 2, replacement of AGCT in CDR3 with TGCT led to significantly decreased SHM (Fig. 3A positions 311G/312C and quantified in Fig. 3B). However, at positions 54G/55C and 89G/90C, TGCT was mutated to the same extent as AGCT (Fig. 3A, rows 2 and 3 compared to row 1, and Fig. 3B). Positions 28C and 243G were also comparably mutated in both AGCT and TGCT contexts. Only positions 27G and 244C underwent significantly reduced SHM in the TGCT context (Fig. 3A, rows 2 and 3 compared to row 1, and Fig. 3B). This suggests that, in many sequence contexts, TGCT and AGCT may be equally mutable (Fig. 3A, rows 2 and 3 compared to row 1).

In comparison, the effects of the AACT substitutions were more severe, with all five exchanges resulting in a significant reduction of mutagenesis. Most strikingly, mutagenesis of AACT at positions 28C, 55C, 90C, and 244C was virtually abolished compared to the same positions in AGCT in wt cells (Fig. 3A, rows 4 and 5 compared to row 1). Thus, AID activity at AGCT versus AACT motifs appears to be underpinned by distinct and incompatible sequence contexts. However, the results also suggest that the sequence locale of CDR3 in B1-8^hi may be special since it was only in this location (311G/312C) that the AACT mutation was not eliminated, albeit very strongly and significantly reduced compared to AGCT in wt cells (Fig. 3A, rows 4 and 5 compared to row 1, and Fig. 3B).

Combining the results in Figs. 2 and 3 leads to the conclusion that sequence contexts favoring mutation of some WRCH motifs may not be fully compatible with mutation of other motifs, and in some cases, such as AGCT versus AACT, may even be incompatible. The results further support the notion that a motif-specific sequence grammar may regulate SHM patterns rather than a universal grammar for all WRCH motifs.

## Motif-context relationships may result from differences in the modes and strengths of ssDNA substrate binding to AID

To investigate whether the changes in sequence context or central motif observed experimentally could result from the structural differences in the binding of AID to different ssDNA sequences, we performed molecular dynamics (MD) simulations (Yoo et al, 2020). We studied ssDNA-protein complexes using ssDNA substrates containing the AGCT motif within the strong B1-8^hi CDR3 context (TACGGTAGTAGCTACTTTGACT) and the weak FWR3 context

(CCTACATGCAGCTCAGCAGCCT). We refer to these substrates as AGCT^strong and AGCT^weak, respectively. We also analyzed binding of ssDNA substrates containing central AACT and TGCA motifs within the same, strong CDR3 context (TACGGTAGTAACTACTTTGACT and TACGGTAGTTGCAACTTTGACT). We refer to these substrates as AACT^strong and TGCA^strong, respectively. The MD simulations were focused on asking whether there was a difference in how the various ssDNA substrates bound to AID.

Binding interactions can be quantified by the free energy of binding, that is, the free energy difference between the ssDNA-protein complex and the unbound ssDNA and protein. As these states differ only in the DNA sequence, the main contribution to the free energy of binding that differs between the states is the enthalpy of the formed complex. An approximation of this contribution is the linear interaction energy (LIE) (Aqvist et al, 1994), which can be computed from MD simulation trajectories. The LIE describes how the interactions between atoms vary when the distance between them is changed, that is, when they are moved away from each other or into closer proximity. A low LIE value corresponds to stronger interactions, while higher values (less negative) are indicative of weaker interactions. There are two main components to this energy: (1) the electrostatic contribution based on interactions between charged regions of the molecules, such as the phosphate backbone of the DNA and the positively charged arginine (Arg) and lysine side chains in the protein, and (2) the van der Waals contribution resulting from other non-electrostatic, non-covalent interactions. The simulations reveal a significantly stronger interaction of AID with AGCT^strong when compared to the other three sequences, while similar interaction strengths were observed for AGCT^weak, AACT^strong, and TGCA^strong (Fig. 4A). This difference is mainly driven by electrostatic interactions, hinting at the importance of Arg side chains in AID in the binding efficiency of ssDNA substrates (Fig. 4A).

To better understand the molecular basis for these differences and their potential impact on deamination, we divided the analysis into context-dependent and motif-dependent binding. To this end, we used two structural measures: a distance measure, which captures how the DNA binds across the protein surface (Fig. 4B) and the root mean square fluctuation (RMSF), which shows the positional fluctuation for each nucleotide and amino acid residue (Fig. 4C). In the distance measure, the smallest variation is seen around position 12, which corresponds to the cytosine in the catalytic pocket (Fig. 4B). In all cases, the highest flexibility was observed at the ends of the ssDNA.

MD reveals a clear change in binding pattern between DNA having different central motifs (Fig. 4B, top panel) and DNA with context changes (Fig. 4B, bottom panel). The changes lie in the interaction of the 5' end of AGCT^strong substrate with the positively charged (Arg-rich) assistant patch (spanning residues 170–178

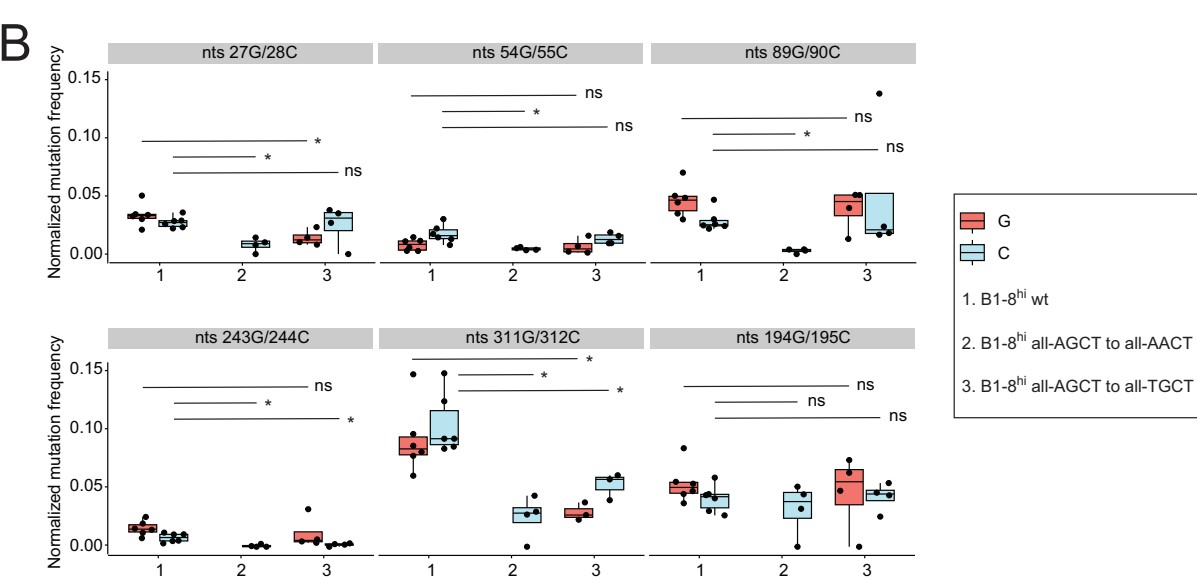

◄ **Figure 3. Sequence contexts supporting AGCT mutagenesis are suppressive for AACT.**

(A) Mutational profiles as in Fig. 2A from B1-8$^{hi}$ wt (row 1) and two clones each of variants wherein all five AGCT residues were replaced by TGCT (allAGCT to allTGCT, rows 2–3) or AACT (allAGCT to allAACT, rows 4–5). The nucleotide positions of the mutated residues are indicated in the top row: 27G/28C, 54G/55C, 89G/90C, 243G/244C and 311G/312C. (B) Box plots quantifying mutation frequencies from six B1-8$^{hi}$ wt samples ($N = 6$) and four samples from all other clones ($N = 4$). Note that for AACT, only 27C, 55C, 90C, 244C, and 312C are AID-target residues, whereas for TGCT, which is a WRCH motif in either orientation, both G and C residues within the motif undergo SHM. The asterisks indicate $P < 0.05$ using the unpaired Student's $t$-test. ns not significant. Box plots were generated as described in Fig. EV1. The exact $P$ values are shown in Dataset EV1.

located within the α6 helix of AID) (Qiao et al, 2017) and surrounding Arg residues (Fig. 4C). AGCT$^{weak}$ does not interact with the assistant patch, reducing potential interactions significantly (Fig. 4C). Thus, the poorer SHM of AGCT$^{weak}$ can be attributed to weaker binding of its DNA context with the assistant patch, whose Arg residues (R171, R174, and R178) were previously shown to be important for mutagenesis on structured DNA substrates (Qiao et al, 2017) and for SHM and CSR (Methot et al, 2018; Qiao et al, 2017). Moreover, since there are no PyPy dimers upstream of the AGCT motif in AGCT$^{strong}$ compared to two overlapping ones in AGCT$^{weak}$, their stark differences in binding to the assistant patch cannot be attributed to PyPy density.

Similar to the distance measure, the RMSF analyses show that for AGCT$^{weak}$ there is reduced flexibility in the N-terminal region of AID around Arg50 (Fig. 4D, bottom panel). This residue is adjacent to the catalytic pocket, lying within a loop between the α2 helix and β2 sheet, and is important for deamination (Qiao et al, 2017). In contrast, the stronger context across the three central motifs shows lower fluctuations in the protein at the C-terminal region containing the assistant patch, suggesting that all three contexts interact comparably with the assistant patch (Fig. 4D, bottom). A corresponding trend is found in the DNA, where the AGCT$^{strong}$ context shows lower fluctuations in the 5' region, which interacts with the assistant patch, while the 3' end is more flexible than the AGCT$^{weak}$ context (Fig. 4D, top panel).

The DNAs with different motifs all bind in similar modes to AID (Fig. 4B,D). However, the binding of the 3' end of the DNA is significantly weaker in the AACT$^{strong}$ and TGCA$^{strong}$ structures than the AGCT$^{strong}$ (Fig. 4D,E). This difference is based on the changes in the interactions of the motif with the surrounding DNA and the binding pocket (Fig. 4F). For AGCT$^{strong}$, Arg25 interacts with nucleotides on either side of the cytosine inside the catalytic pocket, and intra-DNA contacts are formed between the motif and its context (Fig. 4F). This results in interactions of the 3' end with the Arg50 loop. For AACT$^{strong}$ and TGCA$^{strong}$, Arg25 interactions are weakened due to alterations in the interactions with the nucleotides around the motif, such that the local DNA structure at the 3' end becomes more flexible than for AGCT$^{strong}$ (Fig. 4F) and is the likely cause for weaker binding (Fig. 4A,B,E).

Although there appears to be relatively stronger binding (or less flexibility) of the 3' end of DNA to the loop surrounding the catalytic site (including Arg50) for the weak context, it is unable to compensate for the absence of interactions of the 5' end with the assistant patch. However, the analysis of ssDNAs having identical sequence contexts but different central motifs revealed that the 5' ends of all substrates interacted with the assistant patch similarly, but that the 3' ends of the AACT$^{strong}$ and TGCA$^{strong}$ substrates were much more flexible than AGCT$^{strong}$, resulting in weaker interactions with the Arg50 loop.

In sum, the results reveal that changes in sequence context impact assistant patch binding, providing an explanation for the higher mutation rate of AGCT$^{strong}$ compared to AGCT$^{weak}$ (Fig. 2). In contrast, changes in central motifs seem to affect mutagenesis by altering the flexibility of the 3' ends and their interaction with the Arg50 loop. This suggests that the empirically observed relationship between sequence context and WRCH motifs, that is, the motif-context relationship (Fig. 2), may be the result of their differential roles in determining how ssDNA binds to AID. In other words, the binding of ssDNA substrates to the assistant patch via their 5' ends and to the Arg50 loop via their 3' ends are both important, and weakened interaction at either end compromises mutagenesis.

## Motif position significantly influences mutation frequency

Although the above results reveal the role of sequence contexts and motif identity in regulating the differential frequency of mutation at WRCH motifs, they do not rule out the possibility that the position of the motifs within the V region influences SHM frequency. For example, in B1-8$^{hi}$, AGCT$^{FWR3}$ (243G/244C) is mutated 5–10-fold less than AGCT$^{CDR3}$ located 68 bp downstream (311G/312C). This raises the question of whether these positional differences in SHM are due to the sequence contexts of these motifs, their location or a combination of both features.

To test this, we asked whether the weak sequence context of AGCT$^{FWR3}$ would reduce the mutability of AGCT$^{CDR3}$ and, conversely, whether the strong AGCT$^{CDR3}$ context would boost the mutability of AGCT$^{FWR3}$. We generated cell lines where we replaced ±3 bp, ±6 bp, and ±9 bp flanking AGCT$^{CDR3}$ of B1-8$^{hi}$ with the corresponding ±3 bp, ±6 bp, and ±9 bp, respectively, flanking AGCT$^{FWR3}$ (Fig. 5). In effect, this strategy created two identical stretches of AGCT-centered sequences located 68 bp apart within the same V region, spanning 10 bp (3 bp-AGCT-3 bp; ±3 bp), 16 bp (6 bp-AGCT-6 bp; ±6 bp) or 22 bp (9 bp-AGCT-9 bp; ±9 bp). In all these cell lines, V region nascent transcription was similar to that in the Ramos B1-8$^{hi}$ cells (Fig. EV2).

We observed that mutagenesis at AGCT$^{CDR3}$ was decreased at both 311G and 312C in ±3 bp, ±6 bp and ±9 bp clones relative to B1-8$^{hi}$ wt cells (Figs. 5 and 6A). The decrease at 311G was relatively minor and not statistically significant in all cases whereas the decrease at 312C was greater and significant in all cases (Figs. 5 and 6A). However, despite the significant reduction at 312C in the ±3 bp, ±6 bp and ±9 bp clones relative to B1-8$^{hi}$ wt cells, the mutation rates at this residue were still considerably (~4-fold) higher than at the corresponding residue (244C) in AGCT$^{FWR3}$ (Fig. 6A). Importantly, there were no major differences in mutation frequency at either 311G or 312C between the ±3 bp, ±6 bp, and ±9 bp clones, implying that, relative to ±3 bp clones, the additional

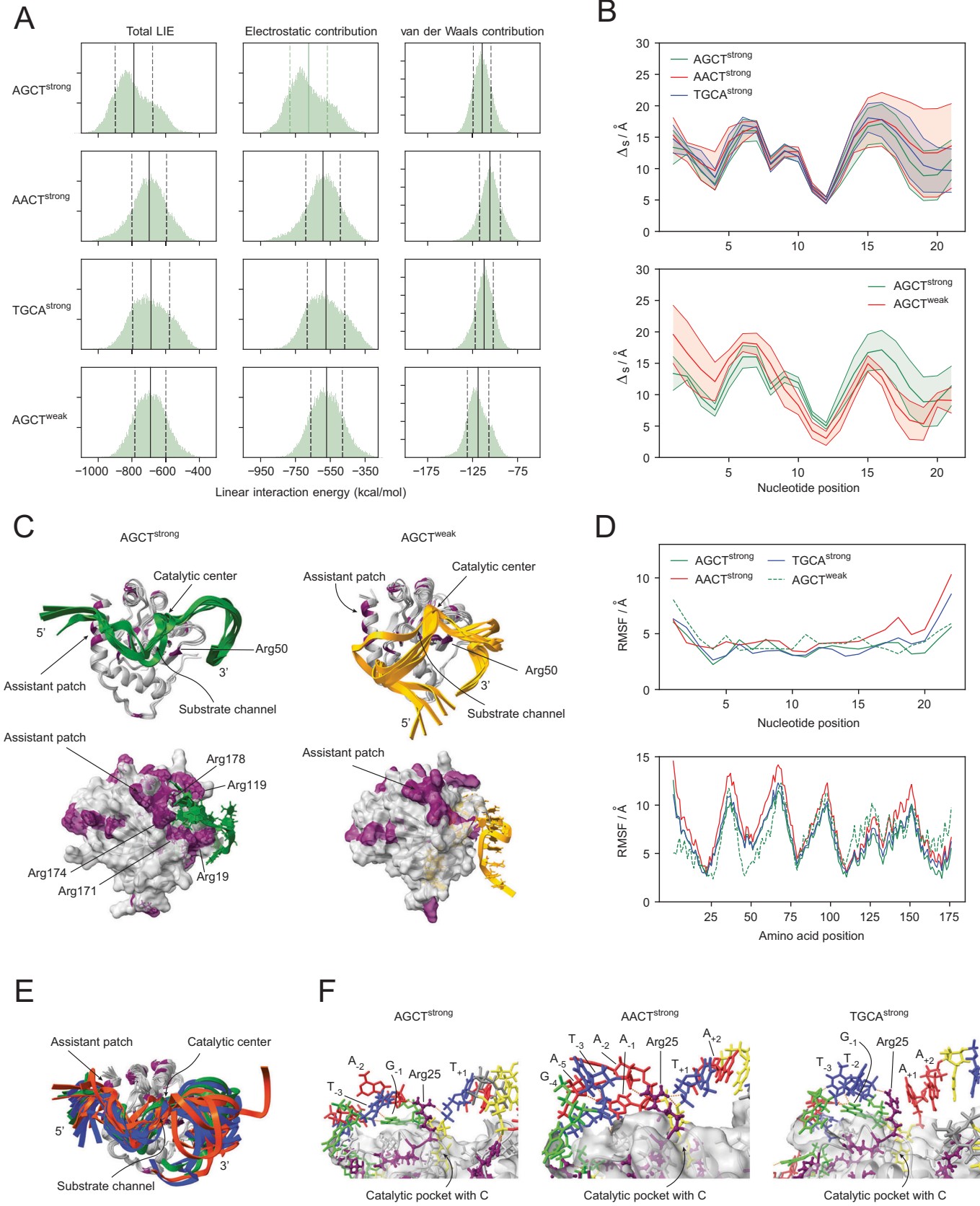

**Figure 4.  Molecular dynamics (MD) simulations evaluating the differential modes and strengths of AID-ssDNA binding upon changes in sequence context or the central motif.**

(A) Histograms of the linear interactions energy (LIE) and the electrostatic and van der Waals contributions for all structures encountered in the MD simulations. The average is shown as a solid line, with the standard deviation shown as dashed lines. AGCT$^{strong}$ (top) shows a lower overall LIE and a lower electrostatic contribution, indicating stronger interactions between this DNA sequence compared to AACT$^{strong}$, TGCA$^{strong}$ and AGCT$^{weak}$. (B) The distance measure ($\Delta_s$/Å) for AGCT$^{strong}$, AACT$^{strong}$ and TGCA$^{strong}$ (top) and for AGCT$^{strong}$ and AGCT$^{weak}$ (bottom). This metric reflects the approximate distance between the surface of the protein and the phosphate group in each nucleotide. The colored range indicates the standard deviation from the average values (thick lines) for each nucleotide. (C) Top: Overlays of structures with strong substrate interactions for AGCT$^{strong}$ (left) and AGCT$^{weak}$ (right) showing the different binding modes. The ssDNA is shown in green (left) and yellow (right). Bottom: Surface plots of the protein with positively charged Arg residues highlighted in purple illustrating how AGCT$^{strong}$ interacts with the assistant patch while this interaction is absent in AGCT$^{weak}$. Arg residues interacting with the ssDNA in AGCT$^{strong}$ are labeled. (D) Root mean square fluctuation (RMSF) for the DNA (top) and protein (bottom) within structures containing the four indicated sequences. Higher fluctuations are seen at the 3′ end of the AACT$^{strong}$ and TGCA$^{strong}$ ssDNA substrates and at the 5′ end of the AGCT$^{weak}$ substrate. In the protein, fewer fluctuations are observed in the loop around residue Arg50 for AGCT$^{weak}$, indicating stronger DNA–protein interactions in this region, while the motifs in the stronger context show lower fluctuations in the C-terminal portion which contains the assistant patch. (E) Aligned structures with average binding showing the lower fluctuation in the 3′ end for the AGCT$^{strong}$ (green) substrate compared to AACT$^{strong}$ (orange) and TGCA$^{strong}$ (blue) substrates. Note that these are a random selection of structures illustrating the variability in the 3′ end seen in the data in (B, D). (F) Interactions of AGCT$^{strong}$ substrate (left) showing strong intra-DNA interactions as well as contact with Arg25 near the active site. By contrast, the presence of AACT (middle) and TGCA (right) central motifs within the same strong context leads to the loss of these interactions. A, G, C and T nucleotides are color-coded in red, green, blue and yellow, respectively.

---

FWR3 sequence context in the ±6 bp and ±9 bp clones had no further impact on the mutation rate of AGCT$^{CDR3}$ (Figs. 5 and 6A). These results suggest that the location of the AGCT motif within the V region is an important determinant of mutability.

Next, we performed the converse experiment wherein we replaced ±3 bp and ±6 bp flanking AGCT$^{FWR3}$ with the corresponding ±3 bp and ±6 bp, respectively, flanking AGCT$^{CDR3}$ (Fig. 6B). There were no significant changes in V region nascent transcription in these new lines compared to Ramos B1-8$^{hi}$ cells (Fig. EV1). Within AGCT$^{FWR3}$, only 244C underwent a statistically significant increase in SHM, whereas 243G showed no significant change in mutability (Fig. 6B,C). Importantly, however, the overall increase in mutability at 244 C was relatively minor, with the increased mutation rate being substantially lower than at 312C in AGCT$^{CDR3}$ (Fig. 6B,C). Moreover, no major changes in AGCT$^{FWR3}$ mutation frequency were observed between the ±3 bp and ±6 bp clones, implying that the additional CDR3 sequence context in the ±6 bp clones did not further increase the mutation frequency of AGCT$^{FWR3}$ in these cells (Fig. 6B,C).

These results show that exchanging the sequence context of AGCT$^{FWR3}$ for that of AGCT$^{CDR3}$, and vice versa, had relatively minor effects on the normal positional differences in SHM between AGCT$^{FWR3}$ and AGCT$^{CDR3}$, respectively. We conclude, therefore, that in addition to sequence context, the position of a WRCH motif within the V region also influences its mutation rate.

### Sequence contexts can regulate the relative mutagenesis of adjacent AID-targeted cytosines within AGCT motifs

A consequence of the above-mentioned changes was a distinct alteration in the relative profile of G versus C mutations within AGCT$^{CDR3}$ and AGCT$^{FWR3}$. In wt cells, 312 C is typically more mutated than 311 G (Figs. 5 and 6A), whereas 244C is typically less mutated than 243G (Fig. 6B,C). However, when AGCT$^{CDR3}$ was flanked by AGCT$^{FWR3}$ sequences, 311G was more mutated than 312 C (Figs. 5 and 6A). Similarly, at AGCT$^{FWR3}$, 244 C was more mutated than 243 G when flanked by CDR3 sequences (Fig. 6B,C). With respect to the direction of sense transcription, these results imply that mutations on the non-template strand (244C and 312C) were significantly affected by changes in sequence context, whereas mutations on the template strand (cytosines paired with 243G and 311G) were not.

We conclude that the relative frequency of deamination at juxtaposed cytosines within the palindromic AGCT motif is also regulated, in part, by sequence context. These findings suggest that AID activity on the template versus non-template strands of the same motif during transcription is also regulated in a sequence context-dependent manner. Therefore, these observations reveal an additional regulatory layer in the mechanism of SHM dictated by local sequence contexts.

## Discussion

Our study reveals that SHM patterns arise from at least two distinct mechanisms: a motif-specific sequence grammar and the position of the motif within the V region. Our analysis of human V region sequences shows that neither the mutability nor the mutation rates of most WRCH motifs correlate with PyPy richness in their immediate neighborhood, and even shows an inverse correlation with PyPy density for some motifs, such as AACT. Thus, although PyPy richness may contribute to the probability of mutation at some AGCT motifs, as shown previously (Wang et al, 2023) (Fig. 1B), it may not have a major impact on the mutability of other WRCH motifs, suggesting that other mechanisms are involved in shaping the observed mutation profiles. Indeed, through a series of experiments, we demonstrate that a sequence context favoring AGCT mutagenesis did not support high rates of mutagenesis of other motifs placed in the same location. A particularly striking result is that the sequence contexts favoring AGCT mutagenesis can be inhibitory for deamination of AACT even though these motifs differ by only one nucleotide. The major inference from these results is that mutability of different WRCH motifs may be regulated, in part, by a motif-specific sequence grammar rather than a motif-agnostic grammar, and that this grammar can promote or dampen mutagenesis.

The precise trajectory of ssDNA in AID's substrate channel has not been experimentally determined. The available crystal structure of AID contains only a single cytosine in the active site and, moreover, lacks the C-terminal helix (amino acids 182–198) (Qiao et al, 2017). Hence, we have used MD simulations to provide insights into what these trajectories may look like and whether they can explain the observed differences in mutability upon changing central motifs or sequence contexts. In this regard, MD simulations

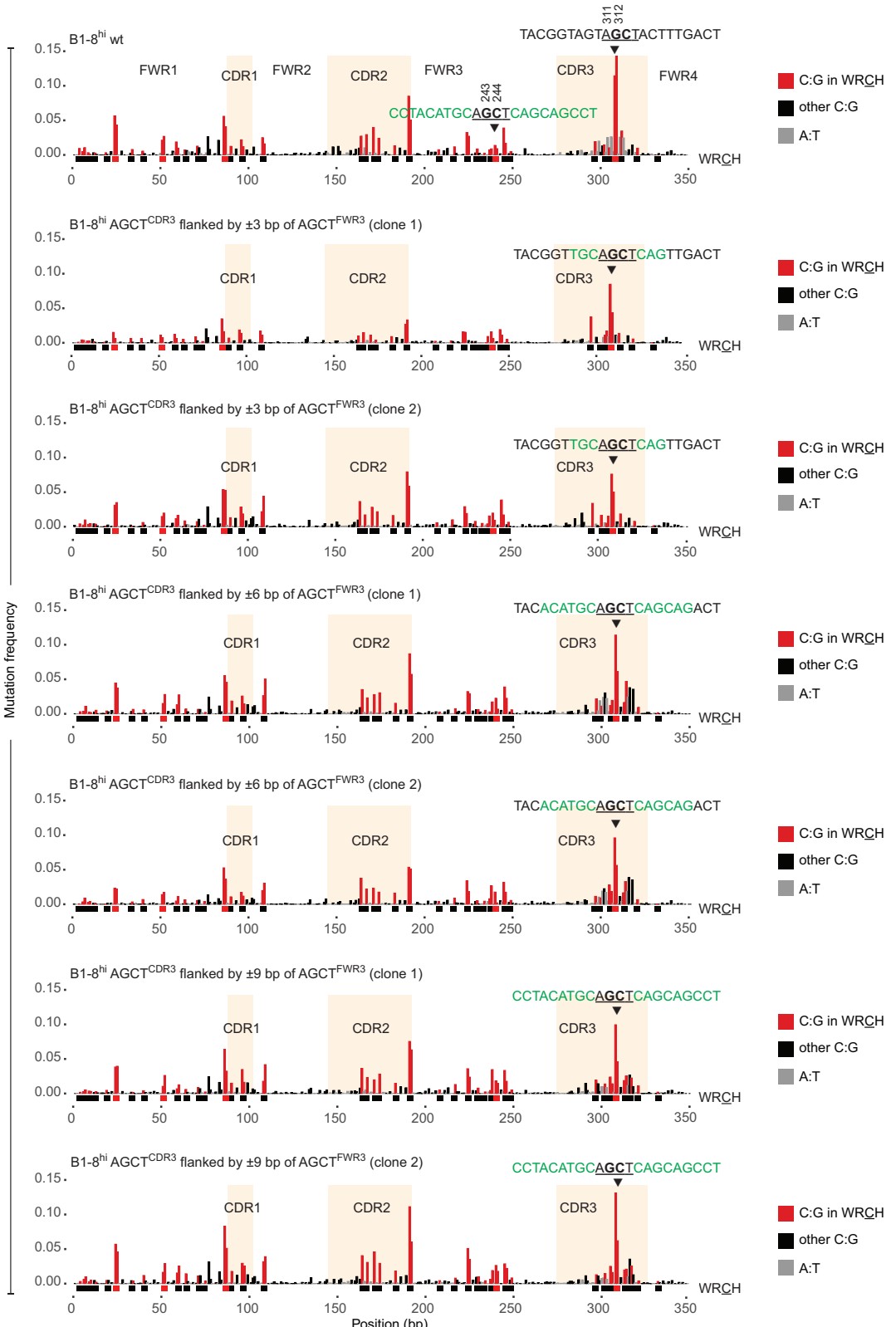

**Figure 5. Evidence that motif location is a major determinant of AGCT mutation rate.**

Mutation profiles of the B1-8^hi V region as in Fig. 2A following the replacement of the sequence locales of AGCT^CDR3 with that of AGCT^FWR3. Sequences flanking AGCT^CDR3 in B1-8^hi wt (top row) were replaced with sequences flanking AGCT^FWR3 (highlighted in green in the top row). The nucleotides from FWR3 flanking AGCT^CDR3 in rows 2–3 (±3 bp), rows 4–5 (±6 bp) and rows 6–7 (±9 bp) are indicated in green.

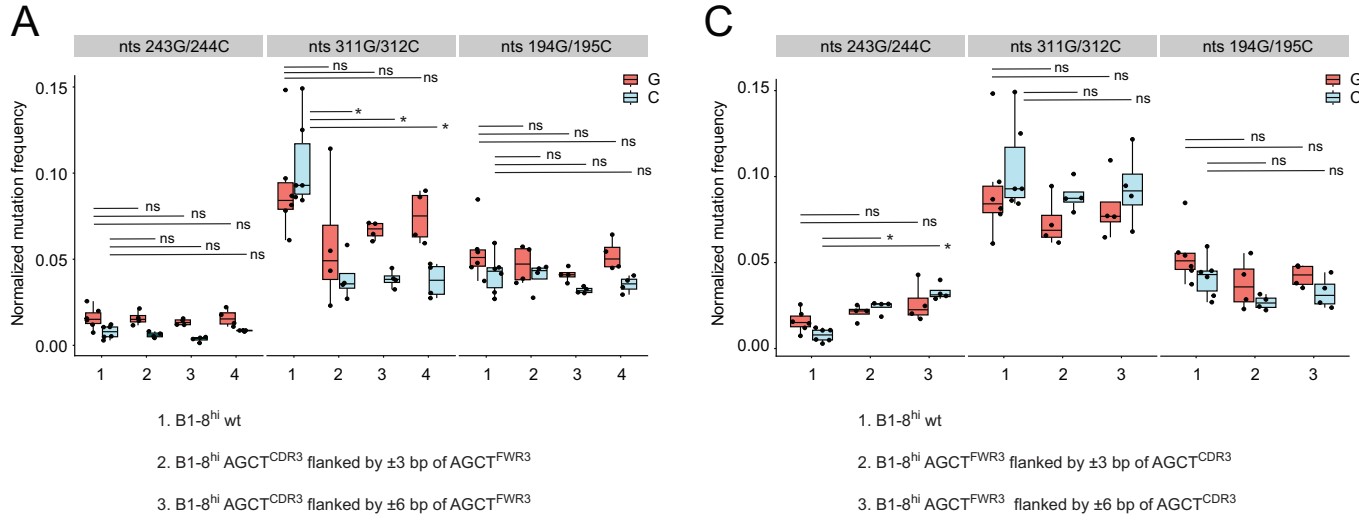

A

B

C

nts 243G/244C    nts 311G/312C    nts 194G/195C

1. B1-8hi wt

2. B1-8hi AGCTCDR3 flanked by ±3 bp of AGCTFWR3

3. B1-8hi AGCTCDR3 flanked by ±6 bp of AGCTFWR3

4. B1-8hi AGCTCDR3 flanked by ±9 bp of AGCTFWR3

1. B1-8hi wt

2. B1-8hi AGCTFWR3 flanked by ±3 bp of AGCTCDR3

3. B1-8hi AGCTFWR3 flanked by ±6 bp of AGCTCDR3

◄ **Figure 6.   Further evidence for the importance of motif location towards mutational outcomes at AGCT motifs.**

(A) Box plots quantifying the MutPE-seq data from Fig. 4. Mutation frequencies were calculated for the indicated residues within AGCT[FWR3] (243G/244C), AGCT[FWR3] (311G/312C) and AGCA[CDR2] (194G/195C). The data represents six samples of B1-8hi wt (*N* = 6) and four samples for all other clones (*N* = 4) The asterisks indicate *P* < 0.05 using the unpaired Student's *t*-test. ns not significant. Box plots were generated as described in Fig. EV1. The exact *P* values are shown in Dataset EV1. (B) Mutation profiles of the B1-8[hi] V region wherein sequences flanking AGCT[FWR3] in B1-8[hi] wt (top row) were replaced with sequences flanking AGCT[CDR3] (highlighted in green in the top row). The nucleotides from CDR3 flanking AGCT[FWR3] in the rows 2–3 (±3 bp) and rows 4–5 (±6 bp) are indicated in green. (C) Box plots representing the data in B. The data represent six samples of B1-8[hi] wt (*N* = 6) and four samples for all other clones (*N* = 4). The asterisks indicate *P* < 0.05 using the unpaired Student's *t*-test. ns, not significant. Box plots were generated as described in Fig. EV1. The exact *P* values are shown in Dataset EV1.

highlighted a dual requirement for robust AID activity: strong interactions at the 5' ssDNA end with the AID assistant patch, and stable engagement of the 3' ssDNA end with the Arg50 loop. Weakened interactions at either end, due to either context-dependent assistant patch binding or motif-dependent 3' end flexibility, directly correlate with decreased mutagenesis. Thus, understanding AID's substrate recognition mechanisms serves as a foundation for predicting mutational hotspots and rationally designing strategies to modulate AID activity.

Another major insight was gained from our finding that the normal positional differences in mutability between strong and weak AGCT motifs were largely retained despite providing the strong AGCT with the sequence context of the weak AGCT, and vice versa. This striking result argues that motif context alone may not always be predictive of mutability or mutation rate, and that the position of the motif can independently and majorly contribute towards mutational outcomes. We note that although the most strongly mutated motif in the B1-8 V region occurs in CDR3, this is not the case for V regions in general. For instance, CDR2 harbors the most mutated residue in VH3-23 (Wei et al, 2015), CDR1 in VH4-59 and VH3-30 (Schoeberl et al, 2025a) and FWR1 in the endogenous Ramos V region, VH4-34 (Schoeberl et al, 2025a). The implication here is that the distance of a motif from the transcription start site is not a major factor in the preferential mutagenesis of some WRCH motifs over others.

Based on these results, our overall conclusion is that the discrete profiles of SHM arise from a combination of motif-specific sequence grammar and motif position. Our work reveals the complexity underlying the origin of SHM patterns and demonstrates that a better understanding of SHM will necessitate deciphering the optimal sequence contexts driving SHM at each WRCH motif and the underlying mechanisms that promote or suppress mutagenesis.

# Methods

## Cell culture

Ramos were cultured in complete RPMI medium (in-house) supplemented with 10% fetal bovine serum (FBS; Invitrogen), glutamine (Invitrogen), sodium pyruvate (Invitrogen), HEPES (in-house), antibiotic/antimycotic (Invitrogen), and β-mercaptoethanol (Sigma). LentiX packaging cells were cultured in complete DMEM medium (in-house) supplemented with sodium pyruvate, 10% fetal bovine serum (FBS; Invitrogen), glutamine (Invitrogen), HEPES (in-house), antibiotic/antimycotic (Invitrogen), and β-mercaptoethanol (Sigma).

## Generation of cell lines

In all cases, the parental cell line was an AID[−/−] IgM[−] Ramos cell clone, which we call VDJ-out, whose generation is described in our

previous study (Schoeberl et al, 2025a), and in which the endogenous V region and promoter were deleted (hence IgM[−]) and replaced with two copies of a unique, GFP-specific small guide RNA (sgRNA)-recognition site. As described (Schoeberl et al, 2025a), the VDJ-out line was targeted by electroporation with the pX458 plasmid (Addgene cat no. 48138) expressing the GFP-specific sgRNA and a homology-directed repair template encoding a new V region under the control of the endogenous Ramos VH4-34 promoter. Electroporation was performed with the NEON NxT electroporation system (Thermo Fisher) in 100 µl of buffer T using the following parameters: 1550 V, 20 ms, 1 pulse. V region sequences were synthesized as gBlocks (IDT), and 5′ and 3′ homology arms were amplified by PCR from wild-type Ramos cell genomic DNA. Five days post-electroporation, single cells were sorted on a BD FACSAria III cell sorter and expanded. Correctly targeted clones were validated by Sanger sequencing. For each V region, two clones were selected for mutational analyses.

## RT-qPCR

RT-qPCR assays with externally spiked-in *Drosophila* S2 cells were described in detail in our previous study (Schoeberl et al, 2025b). *Drosophila* S2 cells and Ramos cells were mixed at a ratio of 1:9 ($0.5 \times 10^5$ S2 and $4.5 \times 10^5$ Ramos cells) followed by total RNA extraction with the TRIzol reagent (Thermo Fisher), DNaseI digestion, and cDNA synthesis with random primers. The $2^{-\Delta\Delta Ct}$ method was used to quantify the data using the *Drosophila Act5c* transcript for normalization.

## Lentiviral infections

Lentiviral pRRL vectors (Dull et al, 1998) expressing AID (m7.3) fused with mCherry (Schoeberl et al, 2025a) were transfected along with ecotropic envelope (Eco-env)-expressing helper plasmid into LentiX cells using polyethylenimine (PEI 25 K, Polysciences). Ramos cells were infected with lentiviral supernatants by spinfection (2350 rpm for 90 min at 37 °C) in the presence of 8 µg/ml polybrene (Sigma). If infection rates were >90%, cells were used directly for MutPE-seq after 21 days. Otherwise, mCherry-positive cells were isolated on a BD FACSAria III cell sorter after 10–12 days and cultured until 21 days, followed by harvesting for MutPE-seq.

## Mutational analysis by paired-end deep sequencing (MutPE-seq)

MutPE-seq was performed as described (Schoeberl et al, 2025a) and was based on the approach described in two previous reports (Robbiani et al, 2015; Yeap et al, 2015) with several modifications. 80 ng of genomic DNA were amplified by PCR with the Kapa HiFi HS 2x RM

(Roche Diagnostics), For the first PCR, we used 20–25 cycles with 0.2 μM locus-specific primers fused to a varying number of random nucleotides and to the first part of the Illumina adapter sequences (FW: 5′-CTCTTTCCCTACACGACGCTCTTCCGATCT-(N)x-gene-specific sequence-3′; RV: 5′-CTGGAGTTCAGACGTGTGCTCTTCCGATCT-gene-specific sequence-3′; see Dataset EV2 for a complete list of primer sequences). Random Ns were used to increase complexity and shift frames of very similar amplicons to improve cluster calling and sample identification. After the first PCR, samples were purified with 0.2x/0.7x SPRI beads (Beckman Coulter), eluted in 10 μl water and amplified for ten cycles with 0.75 μM primers containing linker sequences and dual barcoding: Forward: 5′- AATGATACGGCGACCACCGAGATCTA-CACXXXXXXXXXACACTCTTTCCCTACACGAC-3′ and Reverse: 5′-CAAGCAGAAGACGGCATACGAGATXXXXXXGTGACTGGAGTT-CAGACGTGTGTG-3′. The stretches of X dNTPs (underlined) serve as specific barcodes, creating a unique dual barcode combination for each sample. Alternatively, unique dual index primer pairs (NEB) were used. PCR products were purified either by extraction from a 2% low-melt agarose gel or with 0.7x SPRI beads and eluted in 20 μl water. The quality and concentration of the single libraries were determined on a Fragment Analyzer. Samples were equimolarly pooled for next-generation sequencing on Illumina MiSeq or NextSeq2000 P1 platforms and sequenced paired-end (PE300).

## Analysis of human V regions from BCR-sequencing

We started with pre-processed, non-productive V region sequences kindly provided by Steven Kleinstein, Yale University (Zhou and Kleinstein, 2020) (https://bitbucket.org/kleinstein/projects/src/master/Zhou2020/). The sequences were annotated with IgBLAST 1.21 (Ye et al, 2013). We excluded sequences with unknown ("N") bases, insertions or deletions. Sequences were grouped by their 15-mer or pyrimidine content, depending on the analysis, to calculate mutation frequencies. Groups with a coverage of <30 sequences were excluded from the analysis. Pyrimidine dimer (PyPy) content was calculated by counting all occurrences of two consecutive pyrimidine (C or T) bases in the 15-mer context.

## MutPE-seq data processing

MutPE-seq data were processed as described in detail in our previous study (Schoeberl et al, 2025a). Reads were trimmed for standard adapters with cutadapt (Martin, 2011). Poor quality ($Q < 25$) 3′ bases were trimmed with trimmomatic (Bolger et al, 2014) by averaging over a sliding window of 5nt. Read pairs were then filtered for minimum remaining length (200nt for read 1, 100nt for read 2) using cutadapt. Read mates were merged down to make combined single-end reads with FLASH (Magoc and Salzberg, 2011) allowing 10% mismatch between the mates. Obvious erroneous mergers were removed by selecting combined reads with lengths within ±30nt of the amplicon length using cutadapt. The remaining combined reads were aligned with Bowtie2 (Langmead and Salzberg, 2012), using the "–very-sensitive-local" alignment mode and only the fixed V region sequence and its immediate vicinity as reference. A pile-up was generated with samtools (Li et al, 2009), taking into account only bases with a quality of at least 30. The pileups were then quantified with a custom Python script, and the resulting mutation counts were processed and visualized with custom scripts in R (v3.5.1),

with the help of additional R packages (data.table [https://cran.r-project.org/web/packages/data.table/index.html], ggplot2 [https://cran.r-project.org/web/packages/ggplot2/index.html], ggrepel [https://cran.r-project.org/web/packages/ggrepel/index.html], patchwork [https://cran.r-project.org/web/packages/patchwork/index.html]).

## Calculation of mutation frequency and normalized mutation frequency

We determine the mutation frequency at each nucleotide position by calculating the fraction of mutated bases over the total number of bases covering that position. These values are shown on the Y axis of the mutation frequency histogram (e.g., Fig. 2A). To test for differences between samples, we calculate, for a given position (e.g., 312C), the fraction of mutation frequency at that position over the sum of mutation frequencies in the entire region. These values, which we call normalized mutation frequency, are displayed as box plots (e.g., Fig. 2B). This is necessary because some samples may have an overall higher or lower mutation frequency relative to others. By plotting the mutation measurement per nucleotide as the fraction of the total mutation load, we remove this potential variance, which allows us to compare the differences in mutability at a position of interest (e.g., 312C) between samples.

## Molecular dynamics (MD) simulations

### Rationale for the simulations and their setup

In this study, we focused on the dynamics of the formed complex, that is, the complex of AID bound to substrate ssDNA of different sequences. The rationale for this is as follows. Considering the entropic and enthalpic interactions, the protein component can be ignored as it is identical across all different conditions. The overall change in entropy is similar for any DNA sequence that binds. We see in our simulation data similar scales of mobility of the protein and the DNA in the bound state (that is, similar root mean square fluctuations), and the different ssDNA sequences will have similar entropy when isolated. This similarity in entropy means that any difference in the free energy of binding is based on changes in enthalpy. Again, the various ssDNAs will have very similar enthalpy. Therefore, the difference in binding enthalpy in the different protein-DNA complexes is the most likely driver of potential differences in binding strength, which in turn may impact deamination. Recognition of this single factor as the key driver for potential changes in binding allows us to study the bound protein-DNA complex rather than the process of binding. This significantly reduces the time scales needed for simulations, allowing us to use MD simulations to investigate this problem.

The simulations yield a range of structures and their motions in the bound state. From these sets of structures (known as trajectories), we can identify the average binding mode and interactions, the strongest binding configurations and the interactions that differ between conditions. The simulations need to be initiated from bound structures for the DNA to the protein. However, no such configuration is available from experiments. Instead, we started from a structure of AID in complex with a single cytosine in the active site (PDB ID: 5W0U) (Qiao et al, 2017). We isolated the amino acid residues of AID in this structure, the cytosine and associated ions. Manually, we added

a single nucleotide on either side and optimized the structure to remove clashes between atoms. Then, the 5' and 3' tails were added to give the full structure. Two structures were derived in this fashion, one for AGCT in the strong context (AGCT$^{strong}$) and one in the weak context (AGCT$^{weak}$), which served as starting structures for MD simulations.

In the initial set of simulations, the DNA was allowed to relax and find its binding mode on the surface. The resulting structures were investigated to identify strongly bound complexes. While these already exhibited the difference in binding described in the main text, within each set the low energy structures were well conserved. One structure of these was taken as the starting point for the final simulations. To obtain input structures for the AACT and TGCA motifs, we mutated the AGCT structure and allowed for an additional time window for relaxation.

### Overview of simulations

Table 1 summarizes the simulations used in this study. For AGCT$^{strong}$ and AGCT$^{weak}$, a total of 500 ns each were used for the initial phase and the production phase. For the alternative motifs (AACT$^{strong}$ and TGCA$^{strong}$), a total of 500 ns of MD simulations were run. Though limited in number, the approach underlines the role of these simulations as supportive, initial investigations into whether structure may play a role in the motif-context relationship observed from experiments.

### Simulation protocol

The simulations used AMBER with the OL15 (Galindo-Murillo et al, 2016) force field for DNA, the ff19SB (Tian et al, 2020) force field for the protein and the OPC water model (Izadi et al, 2014) Li/Merz ion parameters (12-6 IOD set) for ions in OPC water were used (Li et al, 2020, 2021; Sengupta et al, 2021). Solvent was added as a truncated octahedron around the formed complex. Cl- ions were added for a concentration of 150 mM, and K+ ions were added subsequently to neutralize the simulation box. The calcium and zinc ions in the original structures were retained. The energy of the solvated system was initially minimized for 5000 steps with harmonic constraints of 50 kcal/mol for all solute atoms to relax the solvent shell. Another 5000 steps followed with no constraints. The system was heated from 0 to 300 K in a 200 ps simulation with harmonic restraints of 50 kcal mol$^{-1}$ and a time step of 2 fs with a Langevin thermostat employing a collision frequency of 0.2 ps$^{-1}$. Three cycles of NVT dynamics of 100 ps each were used to reduce restraints in steps (25, 10, and 5 kcal/mol), followed by 200 ps without any constraints. Pressure was equilibrated for 1 ns with weak constraints (10 kcal/mol) and 1 ns without constraints using a Langevin dynamics with a collision frequency of 1.0 ps−1 and

Berendsen barostat. All simulations were run with pmemd (Case et al, 2023) with GPU support (Götz et al, 2012; Salomon-Ferrer et al, 2013) and setup using AmberTools (https://ambermd.org/AmberTools.php) (Case et al, 2023). The linear interaction energy (LIE) and RMSF were computed using cpptraj (Roe and Cheatham, 2013), which is part of AmberTools (Case et al, 2023).

### Definition of distance measure

Qualifying the mode of binding is challenging as the potential ways the DNA might interact with the protein are numerous. We attempted to find a visualization that gives an idea of the binding across the DNA and the flexibility/variability in the binding. The protein is relatively unchanged during all our simulations (see RMSF plots). We can therefore use the protein as a reference by assuming it to be approximately spherical, using the radius of gyration as the effective sphere size for each structure. Using the reference of the C-alpha atom in Trp79, we compute the distances between the center of the protein and the phosphorus atoms in the DNA. The distance is taken for each phosphorous atom, and the radius of gyration is subtracted. This gives the distance above the approximate sphere for each nucleotide. As the protein is not a perfect sphere and the phosphorous atoms will not necessarily be located right at the surface, the measure is likely larger than zero. It further depends on the protein shape, and changes between DNAs will likely mean the values for each position are not directly comparable. However, the key features that can be compared are: (a) the overall shape of the measure across all nucleotides, and (b) the variation in the measure for each position.

## Statistical analysis

Statistical tests were performed with the unpaired, two-tailed Student's *t*-test, as indicated in the figure legends.

**Reagents and tools table**

| Reagent/resource | Reference or source | Identifier or catalog number |
|---|---|---|
| **Experimental Models** | | |
| Lenti-X cells | Takara Bio | 632180 |
| Ramos VDJout | Schoeberl et al, 2025a | |
| Ramos B1-8hi wt | Schoeberl et al, 2025a | |
| Ramos B1-8hi AGCTCDR3 replaced with AACTCDR3 | This study | |
| Ramos B1-8hi AGCTCDR3 replaced with AGCACDR3 | This study | |
| Ramos B1-8hi AGCTCDR3 replaced with AGCCCDR3 | This study | |
| Ramos B1-8hi AGCTCDR3 replaced with TGCACDR3 | This study | |
| Ramos B1-8hi AGCTCDR3 replaced with TGCTCDR3 | This study | |
| Ramos B1-8hi all-AGCT to all-TGCT | This study | |
| Ramos B1-8hi all-AGCT to all-AACT | This study | |

**Table 1.   Summary of simulations used for molecular modeling.**

| Motif | Context | Purpose | Length | Number of replicas |
|---|---|---|---|---|
| AGCT | Strong | Finding bound structures | 100 ns | 5 |
| AGCT | Weak | Finding bound structures | 100 ns | 5 |
| AGCT | Strong | Production | 100 ns | 5 |
| AGCT | Weak | Production | 100 ns | 5 |
| AACT | Strong | Production | 100 ns | 5 |
| TGCA | Strong | Production | 100 ns | 5 |

| Reagent/resource | Reference or source | Identifier or catalog number |
|---|---|---|
| Ramos B1-8hi AGCTCDR3 ±3 bp of AGCTFWR3 | This study | |
| Ramos B1-8hi AGCTCDR3 ±6 bp of AGCTFWR3 | This study | |
| Ramos B1-8hi AGCTCDR3 ±9 bp of AGCTFWR3 | This study | |
| Ramos B1-8hi AGCTFWR3 ±3 bp of AGCTCDR3 | This study | |
| Ramos B1-8hi AGCTFWR3 ±6 bp of AGCTCDR3 | This study | |
| **Recombinant DNA** | | |
| pX458-sgRNA | Addgene | 48138 |
| pCMVR8.74 | Addgene | 22036 |
| pRRL-SFFV-hAIDm7.3-mcherry | Schoeberl et al, 2025a | |
| pHCMV-EcoEnv | Addgene | 15802 |
| **Antibodies** | | |
| **Oligonucleotides and other sequence-based reagents** | | |
| List of primers, oligos and sgRNAs | This study | Table EV2 |
| Dual index primer pairs | NEB | E6440S, E6442S, E6444S, E6446S |
| **Chemicals, Enzymes and other reagents** | | |
| TriZol | Thermo Fisher | 15596026 |
| Polyethylenimine 25K | Polysciences | 23966 |
| Polybrene | Sigma-Aldrich | 28728-55-4 |
| Kapa HiFi HS 2x RM | Roche Diagnostics | KK2601 |
| SPRI beads | Beckman Coulter | B23318 |
| Superscript RT II | Thermo Fisher | 18064-022 |
| GoTaq RT-qPCR mix | Promega | A6001 |
| RPMI 1640 | in-house | N/A |
| DMEM | in-house | N/A |
| Fetal bovine serum | Gibco | 10270106 |
| HEPES pH 7.3 | in-house | N/A |
| L-Glutamine | Gibco | 25030-024 |
| Sodium pyruvate | Gibco | 11360070 |
| Antibiotic/antimycotic | Fisher Scientific | 15240062 |
| b-Mercaptoethanol | Invitrogen | 21985023 |
| Phenol:Chlorophorm:Isoamyl Alcohol | Invitrogen | 15593049 |
| Glycogen | Thermo Scientific | R0561 |
| DNaseI | New England Biolabs | M0303S |
| UltraPure low-melting Point agarose | Invitrogen | 16520050 |
| Proteinase K | Roche | 03115879001 |
| Q5 High-Fidelity DNA Polymerase | NEB | M0491L |

| Reagent/resource | Reference or source | Identifier or catalog number |
|---|---|---|
| Fragment analyzer HS NGS fragment kit | Agilent Technologies | DNF-474-1000 |
| **Software** | | |
| cutadapt | Martin, 2011 | |
| trimmomatic | Bolger et al, 2014 | |
| FLASH | Magoc & Salzberg, 2011 | |
| Bowtie2 | Langmead & Salzberg, 2012 | |
| samtools | Li et al, 2009 | |
| Data.table | https://cran.r-project.org/web/packages/data.table/index.html | |
| ggplot2 | https://cran.r-project.org/web/packages/ggplot2/index.html | |
| ggrepel | https://cran.r-project.org/web/packages/ggrepel/index.html | |
| patchwork | https://cran.r-project.org/web/packages/patchwork/index.html | |
| pmemd | Case et al, 2023 | |
| GPU support | Götz et al, 2012, Salomon-Ferrer et al, 2013 | |
| AmberTools | https://ambermd.org/AmberTools.php | |
| **Other** | | |
| BD FACS Aria III | BD Biosciences | |
| NEON NxT electroporation system | Thermo Fisher | |
| NextSeq2000 | Illumina | |
| MiSeq2 | Illumina | |
| Fragment analyzer | Agilent Technologies | |

## Data availability

All next-generation sequencing data has been deposited in GEO under accession number GSE283316. Code for the workflow and the custom scripts is available on GitHub at https://github.com/PavriLab/IgH_VDJ_MutPE. MD simulation data has been deposited in Zenodo at https://doi.org/10.5281/zenodo.15756250.

The source data of this paper are collected in the following database record: biostudies:S-SCDT-10_1038-S44318-025-00640-9.

## Peer review information

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

## Acknowledgements

We thank Steven Kleinstein (Yale University, USA) for providing processed V region sequences. We acknowledge the Vienna Biocenter Core Facilities for next-generation sequencing (VBCF-NGS), the IMP/IMBA BioOptics facility for flow cytometry and the IMP/IMBA Molecular Biology Services for Sanger sequencing. The Computational Research, Engineering and Technology Environment (CREATE) at King's College London was used for MD simulations. This work was funded by Boehringer Ingelheim, Österreichische Forschungsförderungsgesellschaft (The Austrian Industrial Research Promotion Agency) Headquarter Grant (FFG-834223), and grants from the Austrian Science Fund to UES (FWF T 795-B30) and RP (FWF P 32043-B). ITS was funded by a grant (no. 23/18320-3) from Fundação de Amparo à Pesquisa do Estado de São Paulo (FAPESP) – The São Paulo Research Foundation. KRV was supported by the Department of Atomic Energy, Government of India, Project Identification No. RTI 4006.

## Author contributions

**Bianca Bartl**: Data curation; Investigation; Methodology; Writing—review and editing. **Ursula E Schoeberl**: Data curation; Formal analysis; Supervision; Investigation; Methodology; Project administration; Funding acquisition; Writing—review and editing. **Renan Valieris**: Data curation; Software; Formal analysis; Investigation; Visualization; Methodology. **Johanna Fitz**: Data curation; Formal analysis; Supervision; Investigation; Methodology; Writing—review and editing. **Konstantin Roeder**: Software; Formal analysis; Investigation; Visualization; Methodology; Writing—review and editing. **Kutti R Vinothkumar**: Formal analysis; Funding acquisition; Investigation; Visualization; Writing—review and editing. **Benjamin Gundinger**: Investigation; Methodology. **Israel Tojal Da Silva**: Resources; Formal analysis; Supervision; Funding acquisition; Project administration. **Rushad Pavri**: Conceptualization; Resources; Formal analysis; Supervision; Funding acquisition; Writing—original draft; Project administration; Writing—review and editing.

Source data underlying figure panels in this paper may have individual authorship assigned. Where available, figure panel/source data authorship is listed in the following database record: .

## Disclosure and competing interests statement

The authors declare no competing interests.

# Expanded View Figures

**Figure EV1.   Analysis of positional differences in mutation frequency of WRCH motifs.**

(**A**) Mutation frequency of the central C within all cytosine-centered 15-mers (left panel) or of the central C within WRCH-centered 15-mers (right panel). The boxes span the interquartile range with the median represented by the horizontal line. The dot size corresponds to the number of unique 15-mers represented by that dot. The numbers of unique 15-mers (*n*) are indicated in brackets. Box plots were plotted with ggplot2 (https://doi.org/10.1007/978-3-319-24277-4). The lower and upper hinges correspond to the first and third quartiles (the 25th and 75th percentiles). The upper whisker extends from the hinge to the largest value no further than 1.5* IQR from the hinge (where IQR is the interquartile range, or distance between the first and third quartiles). The lower whisker extends from the hinge to the smallest value at most 1.5* IQR of the hinge. Data beyond the end of the whiskers are outlier points and are plotted individually. (**B**) Box plots, plotted as above, as above showing the mutation frequency of the central C within the twelve WRCH motif groups that are further categorized by their location within the indicated V sub-regions (FWR1, CDR1, FWR2, CDR2, and FWR3). The values in brackets are the number of unique 15-mers in that group.

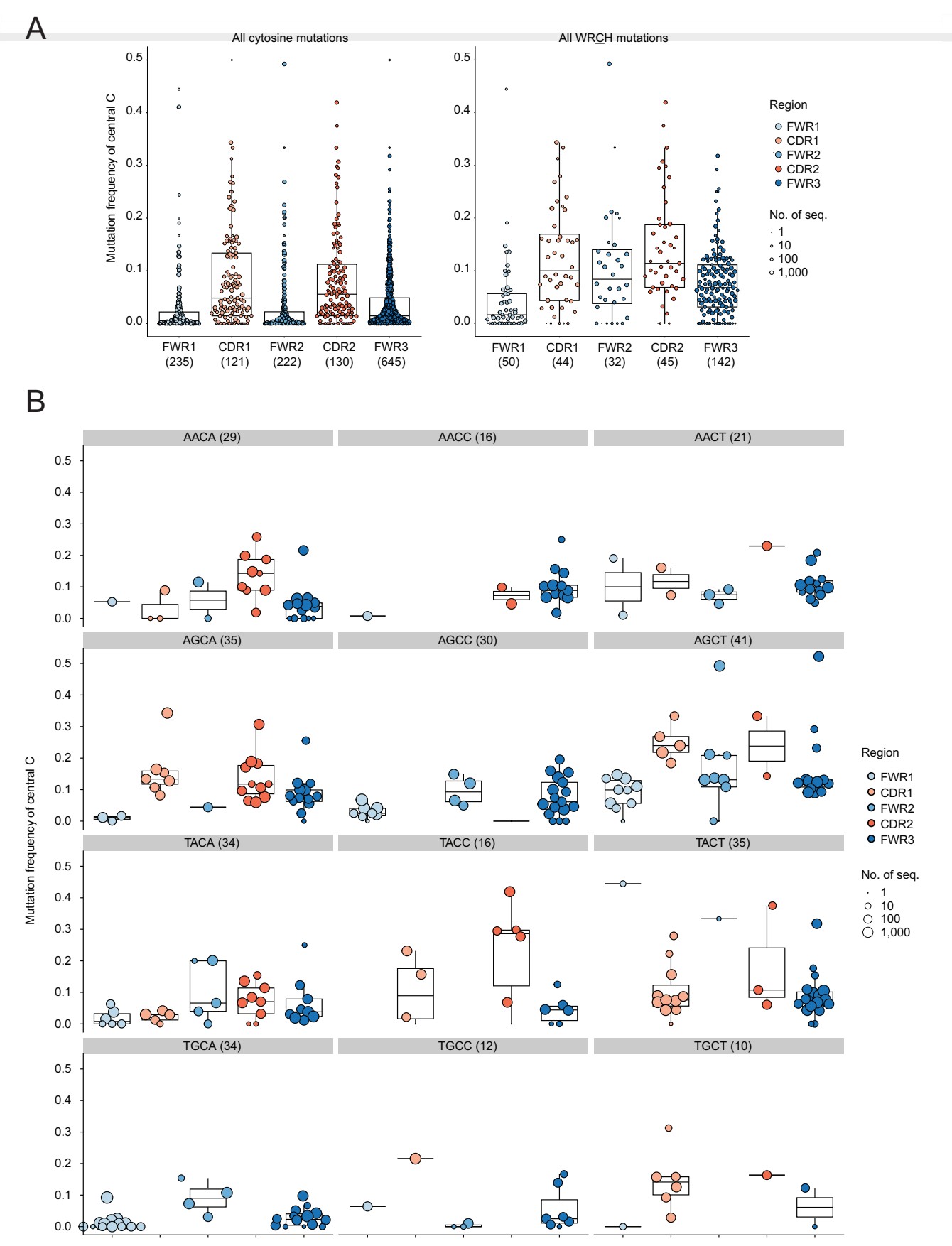

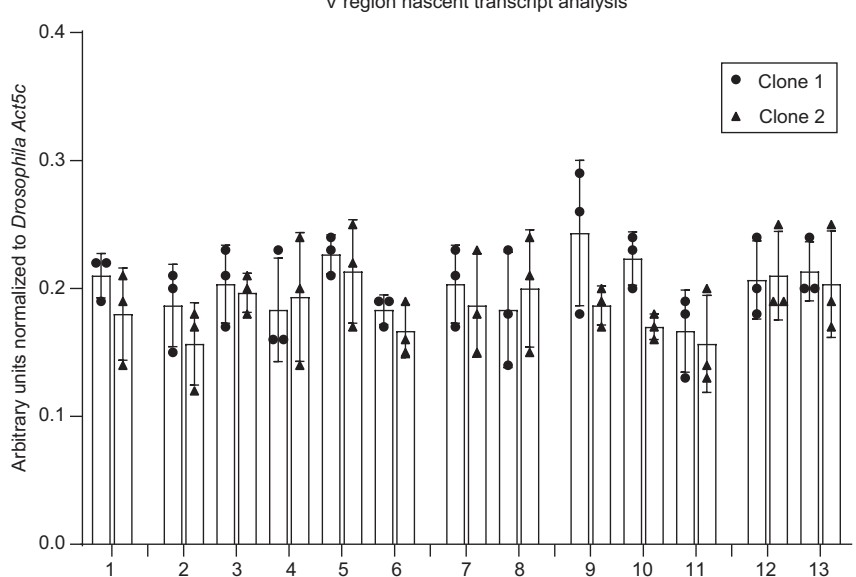

V region nascent transcript analysis

1. B1-8$^{hi}$ wt

2. B1-8$^{hi}$ AGCT$^{CDR3}$ replaced with AACT$^{CDR3}$

3. B1-8$^{hi}$ AGCT$^{CDR3}$ replaced with AGCA$^{CDR3}$

4. B1-8$^{hi}$ AGCT$^{CDR3}$ replaced with AGCC$^{CDR3}$

5. B1-8$^{hi}$ AGCT$^{CDR3}$ replaced with TGCA$^{CDR3}$

6. B1-8$^{hi}$ AGCT$^{CDR3}$ replaced with TGCT$^{CDR3}$

7. B1-8$^{hi}$ all-AGCT to all-AACT

8. B1-8$^{hi}$ all-AGCT to all-TGCT

9. B1-8$^{hi}$ AGCT$^{CDR3}$ flanked by ±3 bp of AGCT$^{FWR3}$

10. B1-8$^{hi}$ AGCT$^{CDR3}$ flanked by ±6 bp of AGCT$^{FWR3}$

11. B1-8$^{hi}$ AGCT$^{CDR3}$ flanked by ±9 bp of AGCT$^{FWR3}$

12. B1-8$^{hi}$ AGCT$^{FWR3}$ flanked by ±3 bp of AGCT$^{CDR3}$

13. B1-8$^{hi}$ AGCT$^{FWR3}$ flanked by ±6 bp of AGCT$^{CDR3}$

**Figure EV2.  Nascent transcription analysis of the V region from all the cell lines used in this study.**

Three replicates from each clone were analyzed ($n = 3$). The error bars show the standard deviation. No significant changes were observed between the Ramos B1-8$^{hi}$ wt cells (column 1) and any other cell line based on the unpaired, two-tailed Student's $t$-test.

