## [Peer Review File · The EMBO Journal]

Somatic hypermutation patterns are shaped by both motif position and sequence grammar

Bianca Bartl, Ursula Schoeberl, Renan Valieris, Johanna Fitz, Konstantin Roeder, Kutti Vinothkumar, Benjamin Gundinger, Israel Tojal da Silva, and Rushad Pavri

Corresponding author(s): Rushad Pavri (rushad.pavri@kcl.ac.uk)

Review Timeline:

Submission Date:	16th Feb 25
Editorial Decision:	25th Mar 25
Revision Received:	24th Jul 25
Editorial Decision:	27th Aug 25
Revision Received:	8th Sep 25
Accepted:	28th Sep 25

Editor: Ioannis Papaioannou

Transaction Report:

Dear Dr. Pavri,

Thank you for submitting your manuscript EMBOJ-2025-120513 for consideration by The EMBO Journal, and for your patience during peer review. Your manuscript has now been seen by three experts in the field, and we have received the full set of their comments, which you can find below.

As you will see, referees #1 and #3 are rather supportive of this work and point out that it reports important findings that should be useful for the field, while the data are convincing and the work well-designed, controlled, and performed. Referee #2, on the other hand, concludes that the depth and significance of this work are not sufficient for publication here, although they recognize that the findings are indeed provocative. All referees make specific and constructive suggestions for the improvement of the work and the manuscript.

On balance, and considering the importance of the topic, the quality of the work, and the positive comments and recommendations of referees #1 and #3, I would like to invite you to submit a thoroughly revised version of your manuscript along with a detailed point-by-point response addressing all referees' comments. I should add that it is The EMBO Journal policy to allow only a single round of major revision, and acceptance of your manuscript will therefore depend on the completeness of your responses in this revised version. Please let me know if you have any questions or comments that you would like to discuss with me. If there are any major points you do not agree with or cannot address during your revision, I would encourage you to share them with me as early as possible to discuss how to proceed further in the most efficient way.

While revising your manuscript, please keep in mind that The EMBO Journal has a very broad readership of molecular biologists working in many different areas of biology. I would therefore like to encourage you to revise your text as appropriate for better accessibility and clarity. The significance and relevance of the findings should be made clear -also to researchers working beyond the specific field- in the revised text, without being overstated, while alternative interpretations and limitations of the work should also be sufficiently discussed, in line also with the referees' comments and suggestions.

We generally allow three months as standard revision time (June 24, 2025). As a matter of policy, competing manuscripts published during this period will not negatively impact our assessment of the conceptual advance presented by your study. However, we request that you contact us as soon as possible upon publication of any related work, to discuss how to proceed. Should you foresee a problem in meeting this three-month deadline, please let us know in advance and we may be able to grant an extension.

Thank you for the opportunity to consider your work for publication in The EMBO Journal. I look forward to your revision.

Best regards,

Ioannis

Instructions for preparing your revised manuscript

1. When you are ready to submit the revision, please upload:

- A Word file of the manuscript text (including legends of main Figures, EV Figures and Tables). Please make sure that changes are highlighted (or "tracked") to be clearly visible.

- Individual production-quality figure files (one file per figure). When assembling your figures, please refer to our figure preparation guidelines in order to ensure proper formatting and readability in print as well as on screen:

If the data shown in a figure are obtained from n {less than or equal to} 2, please use scatter plots showing the individual data points.

i. the name of the statistical test used to generate error bars and P values

ii. the number (n) of independent experiments (please specify technical or biological replicates) underlying each data point (discussion of statistical methodology can be reported in the Materials and Methods section, but figure legends should contain a basic description of n, P, and the test applied)

iii. the nature of the bars and error bars (s.d., s.e.m.).

- A point-by-point response to the referees' comments, with a detailed description of the changes made (as a word file). All referees' concerns must be fully addressed and their suggestions taken on board. When preparing your letter of response to the referees' comments, please bear in mind that this will form part of the Review Process File and will therefore be available online to the community. Please note that you have the possibility to opt out of the transparent process at any stage prior to publication by letting the editorial office know (contact@embojournal.org); if you do opt out, the Review Process File link will point to the following statement: "No Review Process File is available with this article, as the authors have chosen not to make the review process public in this case.". For more details on our Transparent Editorial Process, please visit our website: <https://www.embopress.org/page/journal/14602075/authorguide#transparentprocess>

- Expanded View (EV) files (replacing Supplementary Information) that are collapsible/expandable online. A maximum of 5 EV Figures can be typeset. EV Figures should be cited as "Figure EV1, Figure EV2" etc. in the text, and their respective legends should be included in the manuscript file after the legends of regular figures. See detailed instructions regarding Expanded View files here:

- For the figures that you do NOT wish to display as Expanded View figures, they should be bundled together with their legends in a single PDF file called "Appendix", which should start with a short Table of Contents (including page numbers). Appendix figures should be referred to in the main text as: "Appendix Figure S1, Appendix Figure S2" etc. Please see detailed instructions here: <https://www.embopress.org/page/journal/14602075/authorguide#expandedview>

- A complete author checklist, which you can download from our author guidelines (<https://www.embopress.org/page/journal/14602075/authorguide>). Please note that the checklist will also be part of the Review Process File.

2. Please note that no statistics should be calculated and shown in Figures if n=2. Please also note that each p value should be reported as an exact value.

3. Before submitting your revision, primary datasets (and computer code, where appropriate) produced in this study need to be deposited in appropriate public databases (see <https://www.embopress.org/page/journal/14602075/authorguide#dataavailability>). The accession numbers, database, and the specific URLs (links) should be listed in a formal "Data availability" section (placed after Methods), following the example below:

"The RNA-seq datasets produced in this study are available in the following database:

Gene Expression Omnibus GSE46843 (<https://www.ncbi.nlm.nih.gov/geo/query/acc.cgi?acc=GSE46843>)"

*** All links should resolve to a page where the data can be accessed. ***

*** Please remember to provide in the Data availability section of your revised manuscript reviewer passwords if the datasets are not yet public. ***

*** The Data Availability Section is restricted to new primary data that are part of this study. In case you have no data that require deposition in a public database, please state so instead of referring to the database: "Our study includes no data deposited in public repositories." under the heading "Data availability". ***

4. The materials and methods need to be described in the manuscript using our structured methods format, which is now required for all research articles. According to this format, the Methods section includes a single "Reagents and Tools Table" - listing key reagents, experimental models, software and relevant equipment including their sources and relevant identifiers - followed by a "Methods and Protocols" section describing the methods. Please download and fill our Reagents and Tools Table template (.docx), which you can find in our author guide:

<https://www.embopress.org/page/journal/14602075/authorguide#structuredmethods>. When submitting your revised manuscript, please do not include the Reagents and Tools Table in the Methods section of the manuscript but instead upload it as a separate file choosing the file type "Reagent Table".

5. Please check that the title and the abstract of the manuscript are brief, yet explicit, even to non-specialists. The length of the title should not exceed 100 characters, and the abstract should be a single paragraph not exceeding 175 words.

6. Please also note our reference format: <https://www.embopress.org/page/journal/14602075/authorguide#referencesformat>.

7. At EMBO Press we ask authors to provide source data for the main manuscript figures. Our source data coordinator will contact you to discuss which figure panels we would need source data for and will also provide you with helpful tips on how to upload and organize the files.
8. Please remember: digital image enhancement is acceptable practice, as long as it accurately represents the original data and conforms to community standards. If a figure has been subjected to significant electronic manipulation, this must be noted in the figure legend or in the "Materials and Methods" section. The editors reserve the right to request original versions of figures and the original images that were used to assemble the figure.
9. Our journal encourages inclusion of data citations in the reference list to directly cite datasets that were obtained from public databases. Data citations in the article text are distinct from normal bibliographical citations and should directly link to the database records from which the data can be accessed. In the main text, data citations are formatted as follows: "Data ref: Smith et al, 2001" or "Data ref: NCBI Sequence Read Archive PRJNA342805, 2017". In the Reference list, data citations must be labeled with "[DATASET]". A data reference must provide the database name, accession number/identifiers, and a resolvable link to the landing page from which the data can be accessed at the end of the reference. Further instructions are available at: <https://www.embopress.org/page/journal/14602075/authorguide#referencesformat>.
10. We request authors to consider both actual and perceived competing interests. Please review our policy (<https://www.embopress.org/page/journal/14602075/authorguide#conflictsofinterest>) and update your competing interests statement if necessary. Please name this section 'Disclosure and competing interests statement' and place it after the Acknowledgements section.
11. Please note that all corresponding authors are required to provide an ORCID ID upon submission of a revised manuscript (<https://orcid.org/>). Please find instructions on how to link your ORCID ID to your account in our manuscript tracking system in our Author guidelines (<https://www.embopress.org/page/journal/14602075/authorguide#authorshipguidelines>).
12. We use CRediT to specify the contributions of each author in the journal submission system. CRediT replaces the author contribution section, which should be removed from the manuscript. Please use the free text box to provide more detailed descriptions. See also guide to authors: <https://www.embopress.org/page/journal/14602075/authorguide#authorshipguidelines>.
13. Further information is available in our Guide For Authors: <https://www.embopress.org/page/journal/14602075/authorguide>
14. We would also welcome the submission of cover suggestions or motifs to be used by our Graphics Illustrator in designing a cover.
15. Please use the link below to submit your revision:
<https://emboj.msubmit.net/cgi-bin/main.plex>

Referee #1:

The authors perform an in-depth analysis of the sequence and location determining AID activity at defined WRCH hotspots. A computational analysis of human BCR sequences failed to show any general association between mutation frequency or mutability (except for specific WRCH motifs) and the presence of contextual PyPy, which has been proposed to provide a biochemical advantage to AID.

The authors then construct a set of RAMOS cell lines carrying the B1-8 VDJh, in which the most mutated WRCH motif (AGCT in CDR3) was mutated to 5 other WRCH versions. This enabled analyzing mutability of these motifs at a fixed location. All other motifs, including TGCA, displayed reduced mutability. A similar experiment now changing the context of a given motif by replacing each AGCT hotspots to another motif in their original location, showed changes in mutability indicating that the sequence context ability to increase or decrease the mutability depends on the identity of the motif.

Finally, the authors replaced the sequence context of a CDR3 hot spot for the context of a cold spot WRCH in the FRW3. This enabled to assess the role of DNA context without changing location along the VH. The cold spot was significantly mutated in the new location. Conversely, the context of the hotspot was unable to attract substantial SHM to the cold spot in FWR3. These experiments also revealed a relatively stronger effect of the context on the non-template strand of transcription.

Overall, the results here convincingly show that mutability of a WRCH motif depends on the combination of context and specific motif (hence becoming interdependent variables), with an additional layer defined by their position within the V region.

While the concept of motif mutability, context dependence, and VH position effect of DNA sequence context has been explored in several papers (well referenced here), this manuscript integrates those previous observations into a comprehensive framework that alerts against simpler interpretations based on partial analyses. Importantly, they provide data using a relevant cellular model for SHM and carefully dissect the multiple variables (motif, context, location). The experiments are well designed and controlled. The SHM analyses seem adequate, to the extent that a non-bioinformatician like me can judge.

I don't have major criticisms, but the authors could do some additional analyses and probably discuss some limitations of their work largely done on B1-8 IgH.

One question is whether the positional effect is specific to each VDJ or if there are general rules to be found. For example, the authors find a CDR3 hotspots is the most mutated. But Wei et al (PMID: 25646473, referenced by the authors) found that a hot spot in CDR2 is the most mutated in the endogenous VDJ rearrangement of RAMOS cells. This may be different in other rearrangements. Could the author analyse the extensive data used in figure 1 to determine whether positional effects are rearrangement-specific or if there are some general principles regarding position (e.g. CDR3>CDR2, etc)?

To what extent does the biochemical preferences of AID correlate with the in vivo pattern reported here? The authors' data suggests that the work of Wang et al. (PMID: 37098343, referenced in the paper) may be a special case rather than a general mechanism, which is important to report. However, there is abundant data from the group of Dr Goodman establishing biochemical preferences in V regions transcribed and mutated in vitro. The analysis could benefit of contrasting that data with observed preferences. For example, are biochemical hot or cold spots are within "compatible" contexts or locations? Does the "location" variable act in vitro or only in vivo?

As a minor note, the paper analyzes the mutability of motifs, so it would be more accurate to refer to it as such in abstract and section titles rather than as SHM, which implies the whole process or can be confused with overall mutagenesis.

In this context, Wei et al showed that removing certain hot spots reduced SHM overall in the V regions, a phenomenon that this paper does not observe. This should probably be discussed.

Referee #2:

In this manuscript, the authors examine how local sequence context and position within a V region influence SHM frequencies. They insert different sequences into the position of Vh in Ramos and then deep sequence to measure mutation distribution. They report several interesting observations that argue for this V region in the Ramos cell line using a hyperactive AID mutant, that local sequence context is not sufficient to predict mutation frequencies, that different hotspots behave differently in different positions and contexts, and that local sequence context can influence the relative frequency of mutation at the adjacent GC residues in the center of WGCW motifs. The findings indicate that there is more to learn about how sequence context and position influence SHM profiles. While the findings are provocative, no mechanistic insight is provided and I do not believe this work has sufficient depth and significance for publication in EMBO J.

The work could be improved as described below.

Major comments

1. A few of the key conclusions should be confirmed by expression of WT AID.
2. The analyses in Fig. 1 suffer from the limitation that the mutation frequencies could be significantly influenced by selection. Extrapolating from the previous study with in-frame and out-of-frame B1-8 alleles to conclude that selection can be ignored is not well justified. The conclusions drawn from Fig. 1 are not convincing.
3. The analysis of Fig. 2 is provocative but the authors are comparing absolute mutation frequencies from different cell lines and experiments against one another. There could be variation in overall mutation frequency between experiments, and visually, this appears to be the case for the AGCA experiments, where mutation appears lower across the board. The authors should also plot the data as fraction/percent of the total mutation load, so that such variation is corrected for. This comment applies equally to Figures 3-5 (e.g., the +/-3 clone 1 line of data in Fig. 4 shows reduced mutation across the entire V region).

Minor comments

4. Lines 70-72: Regarding reference 32, I do not feel that it is appropriate to cite a non-peer reviewed article that is almost 3 years old to support a strong claim such as this. Why hasn't this work been published? At the very least, the sentence should be rewritten to express an appropriate degree of doubt that accompanies non-peer reviewed work (best yet would be a clear statement that work has not yet been peer reviewed). This same concern applies to the subsequent statement in the manuscript that cites reference 32 (line 155).
5. Lines 78-79 do not fully capture the scope and relevance of the findings reported in reference 33. Furthermore, the statement on lines 94-96 is somewhat misleading: ref 33 addressed much more than AGCT motifs. Indeed, some of the analyses involved heterogeneous sequences covering many different WGCW motifs.

Referee #3:

A driving question in the field of antibody affinity maturation and somatic hypermutation biology is how does AID identify its target DNA sequence to hypermutate both at V gene sequences (in the Igh) and at other oncogenic DNA sequences elsewhere in the B cell genome. Previous studies have suggested that the presence of two pyrimidine residues (PyPy) 7-15 bps apart from a AID target RGYW sequence significantly promotes AID SHM activity. This was proposed to be due to the ability of this PyPy containing sequence to invade the AID protein minor groove (Assistant patch) (Dai et al, Cell 2023) and place the nearby WRCH sequence near the AID catalytic domain. In this study, the authors carefully investigate this possibility in a Ramos cell line system where a VH4-34 single V gene is replaced with a VHb1-8 allele and expressing an hyperactive mutant of AID. The authors conclude that "human V region sequences shows that neither the mutability nor the mutation rates of most WRCH motifs correlate with PyPy richness in their immediate neighborhood..... Although PyPy richness may contribute to the probability of mutation at some AGCT motifs, it may not have a major impact on SHM of other WRCH motifs". This is an important finding that should be very useful for the field. It is important that alternatives that can explain the discrepancy in the Dai et al paper is discussed properly.

1. Dai et al demonstrated their study in mouse models whereas this study is completely done in Ramos cell lines with over expression of aa hyperactive AID. This difference should be considered and discussed as necessary.
2. One question arises whether the transcription of the B1-8 allele is affected and following the removal of the two Py residues increased transcription compensates for reduction in AID intrinsic SHM activity. Transcription levels of the various mutated versions of B1-8 alleles should be considered.
3. Can the authors discuss how does the AID minor groove binding property of PyPy stand with respect to the their own findings in the discussion section. Does this mean the previous interpretation of the AID structure was not accurate, or does it mean that only PyPy places AGCT (and not other WRCHs) near the catalytic domain of AID. To me it wouldn't make sense that PyPy provides a proper structural context for AGCT SHM, but not for other WRCHs.
4. Authors propose that 312 AGCT SHM is a privileged position due to its distance from the TSS and that makes it mutable, and not the presence of two Py residues. In that case, if the authors increase the distance from TSS by 100 bps by incorporating a random sequence, it should reduce mutational frequency at "312" AGCT? Another questions is, would the sequence leading upto the 312 AGCT matter? This can be discussed in this paper.

Bartl, Schoeberl, Valieris et al. Response to reviewers

We thank the reviewers for their time and for their thorough and insightful feedback and
suggestions. We have edited the manuscript and figures accordingly and added a new analysis
involving molecular dynamics simulations to study the changes in AID-ssDNA binding using
different ssDNA substrates from our mutation analyses.

Below, we address each point in blue text.

**Referee #1:**

The authors perform an in-depth analysis of the sequence and location determining AID activity at
defined WRCH hotspots.

A computational analysis of human BCR sequences failed to show any general association between
mutation frequency or mutability (except for specific WRCH motifs) and the presence of contextual
PyPy, which has been proposed to provide a biochemical advantage to AID.

The authors then construct a set of RAMOS cell lines carrying the B1-8 VDJh, in which the most
mutated WRCH motif (AGCT in CDR3) was mutated to 5 other WRCH versions. This enabled
analyzing mutability of these motifs at a fixed location. All other motifs, including TGCA, displayed
reduced mutability. A similar experiment now changing the context of a given motif by replacing each
AGCT hotspots to another motif in their original location, showed changes in mutability indicating that
the sequence context ability to increase or decrease the mutability depends on the identity of the
motif.

Finally, the authors replaced the sequence context of a CDR3 hot spot for the context of a cold spot
WRCH in the FRW3. This enabled to assess the role of DNA context without changing location along
the VH. The cold spot was significantly mutated in the new location. Conversely, the context of the
hotspot was unable to attract substantial SHM to the cold spot in FWR3. These experiments also
revealed a relatively stronger effect of the context on the non-template strand of transcription.
Overall, the results here convincingly show that mutability of a WRCH motif depends on the
combination of context and specific motif (hence becoming interdependent variables), with an
additional layer defined by their position within the V region.

While the concept of motif mutability, context dependence, and VH position effect of DNA sequence
context has been explored in several papers (well referenced here), this manuscript integrates those
previous observations into a comprehensive framework that alerts against simpler interpretations
based on partial analyses.

Importantly, they provide data using a relevant cellular model for SHM and carefully dissect the
multiple variables (motif, context, location). The experiments are well designed and controlled. The
SHM analyses seem adequate, to the extent that a non-bioinformatician like me can judge.

I don't have major criticisms, but the authors could do some additional analyses and probably discuss
some limitations of their work largely done on B1-8 IgH.

One question is whether the positional effect is specific to each VDJ or if there are general rules to be
found. For example, the authors find a CDR3 hotspots is the most mutated. But Wei et al (PMID:
25646473, referenced by the authors) found that a hot spot in CDR2 is the most mutated in the
endogenous VDJ rearrangement of RAMOS cells. This may be different in other rearrangements.
Could the author analyse the extensive data used in figure 1 to determine whether positional effects
are rearrangement-specific or if there are some general principles regarding position (e.g.
CDR3>CDR2, etc)?

This is a good point, and we have addressed it with a new analysis where we look at mutation
frequency of each WRCH-centered 15-mer within the FWRs and CDRs using the datasets

employed in Fig. 1 (new Fig. EV1). The results show that the overall bias towards CDRs seen in
cumulative analyses arises from the contribution of a few motifs. Even within these, some FWR
15-mers often show similar mutation load as those in CDRs. Thus, based on this analysis, there
are no generally discernible patterns or rules that can explain the position effects.

Please note that Wei et al. mentioned by the reviewer did not analyse the endogenous Ramos V
region but the VH3-23*01 expressed from the *IGH* locus in Ramos cells. They did find that CDR2
harbored the most mutated residue. We have analysed the endogenous V region (VH4-34) in
Ramos cells wherein the most mutated hotspot is in FWR1 (doi.org/10.7554/eLife.106566). We
also analyzed two other human V regions in this study (VH3-30 and VH4-59) wherein the major
hotspots lie in CDR1. We have mentioned this point in the revised Discussion. Thank you for
raising it.

To what extent does the biochemical preferences of AID correlate with the *in vivo* pattern reported
here? The authors' data suggests that the work of Wang et al. (PMID: 37098343, referenced in the
paper) may be a special case rather than a general mechanism, which is important to report.
However, there is abundant data from the group of Dr Goodman establishing biochemical preferences
in V regions transcribed and mutated *in vitro*. The analysis could benefit of contrasting that data with
observed preferences. For example, are biochemical hot or cold spots are within "compatible"
contexts or locations? Does the "location" variable act *in vitro* or only *in vivo*?

For the B1-8 V region used in our work, we note that a biochemical analysis was performed by
Wang et al. (PMID: 37098343) and they noted similarities between the *in vivo* and *in vitro* SHM
patterns in the B1-8 V region. The Goodman lab has done deamination assays with the VH3-23*01
but the raw data has not been deposited and so we cannot make a comparison with the *in vivo*
data. The Wang et al. results from the B1-8 V region would suggest that the biochemical data
recapitulates the relatively higher mutability of at least the major hotspots (such as the AGCT in
CDR3).

As a minor note, the paper analyzes the mutability of motifs, so it would be more accurate to refer to it
as such in abstract and section titles rather than as SHM, which implies the whole process or can be
confused with overall mutagenesis.

We understand and agree with the reviewer's point, and we have reduced the usage of the term
SHM in the main text and section titles.

In this context, Wei et al showed that removing certain hot spots reduced SHM overall in the V
regions, a phenomenon that this paper does not observe. This should probably be discussed.

It is correct that we only observe very localized changes in mutation profiles upon alteration of
specific hotspots or their locales. Though different from Wei et al., it is consistent with several
previous reports in mice and cell lines where changes in sequence context only affected mutation
patterns locally (PMID: 26582132, 28747530 and 37098343).

**Referee #2:**

In this manuscript, the authors examine how local sequence context and position within a V region
 influence SHM frequencies. They insert different sequences into the position of Vh in Ramos and then
 deep sequence to measure mutation distribution. They report several interesting observations that
 argue for this V region in the Ramos cell line using a hyperactive AID mutant, that local sequence
 context is not sufficient to predict mutation frequencies, that different hotspots behave differently in
 different positions and contexts, and that local sequence context can influence the relative frequency
 of mutation at the adjacent GC residues in the center of WGCW motifs. The findings indicate that
 there is more to learn about how sequence context and position influence SHM profiles. While the
 findings are provocative, no mechanistic insight is provided and I do not believe this work has
 sufficient depth and significance for publication in EMBO J.

The work could be improved as described below.

Major comments

1. A few of the key conclusions should be confirmed by expression of WT AID.

Although performing all the experiments with WT AID would be beyond the scope, we provide
 below an experiment using WT AID compared to AIDm7.3 showing that the patterns of mutation at
 the Ramos V region are comparable between them. The reason for choosing AIDm7.3 is that the
 WT AID is much less efficient, requiring 7-8 weeks compared to m7.3 which can be completed in a
 week. We also see the same patterns at this V region using the C-terminally truncated AID mutant
 (JP8Bdel) which is retained in the nucleus due the absence of the nuclear export signal thereby
 generating high mutation frequencies within a few days (please see doi.org/10.7554/eLife.106566).

Furthermore, the levels of AIDm7.3 have no major impact on mutation profiles as seen from the
 analysis below.

Thus, the patterns of SHM are not affected by using hyperactive AID variants or by the levels of
 expression, allowing one to use them for mechanistic studies. In this regard, we note that AIDm7.3
 has been used extensively by David Schatz, an eminent expert in the field, over the past several
 years to discover many important mechanistic features of SHM, including identifying new factors
 by CRISPR screening (e.g. PMID: 40049160, 35450882 and 31821534).

2. The analyses in Fig. 1 suffer from the limitation that the mutation frequencies could be significantly
 influenced by selection. Extrapolating from the previous study with in-frame and out-of-frame B1-8
 alleles to conclude that selection can be ignored is not well justified. The conclusions drawn from Fig.
 1 are not convincing.

Please note that the entire analysis in Fig. 1 (and the new Fig. EV1) was performed with non-
productive BCR sequences which are not subject to selection. We had mentioned this in the
manuscript (first sentence of the Results section and in the Methods). In Ramos cells *in vitro*, there
is no selection occurring since these cells survive without the BCR, hence there is no bias in this
regard.

3. The analysis of Fig. 2 is provocative but the authors are comparing absolute mutation frequencies
from different cell lines and experiments against one another. There could be variation in overall
mutation frequency between experiments, and visually, this appears to be the case for the AGCA
experiments, where mutation appears lower across the board. The authors should also plot the data
as fraction/percent of the total mutation load, so that such variation is corrected for. This comment
applies equally to Figures 3-5 (e.g., the +/-3 clone 1 line of data in Fig. 4 shows reduced mutation
across the entire V region).

We apologise for this error on our part. The bar plots are in fact the normalized mutation frequency
calculated as the reviewer mentions (and now included in the Methods of the revised manuscript).
In essence, we calculate the fraction of mutation frequency at a given position over the sum of
mutation frequency in the entire region. These values are shown on the Y axis of Figures 2B, 3B
and 6A & C. We have re-labelled the Y axes of these box plots as "Normalized mutation
frequency." Thank you for pointing this out.

Minor comments

4. Lines 70-72: Regarding reference 32, I do not feel that it is appropriate to cite a non-peer reviewed
article that is almost 3 years old to support a strong claim such as this. Why hasn't this work been
published? At the very least, the sentence should be rewritten to express an appropriate degree of
doubt that accompanies non-peer reviewed work (best yet would be a clear statement that work has
not yet been peer reviewed). This same concern applies to the subsequent statement in the
manuscript that cites reference 32 (line 155).

This work has been reviewed at eLife and the reviews are available online:
doi.org/10.7554/eLife.106566. As seen therein, all three reviews were very positive with the
reviewers noting the importance of the work for the field, the solid nature of the data and overall
rigor of the study. It has also been cited in a recent review by Uttiya Basu, an eminent researcher
in the field (PMID: 38428317).

5. Lines 78-79 do not fully capture the scope and relevance of the findings reported in reference 33.
Furthermore, the statement on lines 94-96 is somewhat misleading: ref 33 addressed much more
than AGCT motifs. Indeed, some of the analyses involved heterogeneous sequences covering many
different WGCW motifs.

We have changed these lines to incorporate additional findings from the Wang et al study. Please
note that Wang et al. initially did a general analysis of mutability across V regions in different
species using *in vitro* deamination assays with recombinant AID (Figs. 1 & 2 of their paper). This
would have included all motifs, but these analyses were aimed at addressing the evolutionary bias
towards higher mutability of CDRs compared to FWRs. Of note, they did not perform an analysis of
individual WRC motifs from these data. Most importantly, their subsequent analyses (Fig. 3
onwards) focused on AGCT motifs. This included the functional and mechanistic studies *in vitro*
and *in vivo* which formed the basis for their major conclusions regarding the role of PyPy dimers or
Py-richness in promoting mutability. Specifically, the extensive *in vitro* biochemical analyses of Py-
rich sequence contexts (Figs. 4 and 5) and the genetics in mice and murine CH12 cells (Figs. 6)
focused on AGCT-centered motifs. Moreover, the transgenic mice analyzed in Fig. 3 were
generated to harbor sequence context changes relative to AGCT motifs in CDR3 of the B1-8 V

region. Hence, our original statement about this study was based on their core mechanistic
 findings which were derived from AGCT-centered experiments.

**Referee #3:**

A driving question in the field of antibody affinity maturation and somatic hypermutation biology is how
 does AID identify its target DNA sequence to hypermutate both at V gene sequences (in the Igh) and
 at other oncogenic DNA sequences elsewhere in the B cell genome. Previous studies have suggested
 that the presence of two pyrimidine residues (PyPy) 7-15 bps apart from a AID target RGYW
 sequence significantly promotes AID SHM activity. This was proposed to be due to the ability of this
 PyPy containing sequence to invade the AID protein minor groove (Assistant patch) (Dai et al, Cell
 2023 and place the nearby WRCH sequence near the AID catalytic domain. In this study, the authors
 carefully investigate this possibility in a Ramos cell line system where a VH4-34 single V gene is
 replaced with a VHb1-8 allele and expressing an hyperactive mutant of AID. The authors conclude
 that "human V region sequences shows that neither the mutability nor the mutation rates of most
 WRCH motifs correlate with PyPy richness in their immediate neighborhood..... Although PyPy
 richness may contribute to the probability of mutation at some AGCT motifs, it may not have a major
 impact on SHM of other WRCH motifs". This is an important finding that should be very useful for the
 field. It is important that alternatives that can explain the discrepancy in the Dai et al paper is
 discussed properly.

We believe the reviewer is referring to the Wang et. al. Cell 2023 paper (PMID: 37098343). We are
 not aware of a Dai et al. Cell 2023 paper.

1. Dai et al demonstrated their study in mouse models whereas this study is completely done in
 Ramos cell lines with over expression of aa hyperactive AID. This difference should be considered
 and discussed as necessary.

Please also see our response to reviewer 2 where we show MutPE-seq data comparing WT
 versus AIDm7.3 and different expression levels of AIDm7.3 (lines 107-131).

In addition, the patterns of mutations at C:G residues in the B1-8 V region are comparable
 between Ramos cells expressing AIDm7.3 and murine germinal center B cells, as shown below

(taken from our recent work: doi.org/10.7554/eLife.106566).

Collectively, this points to the fact that the discrete mutation profiles are the result of sequence
intrinsic mechanisms (i.e. sequence contexts) rather than the levels or activity of AID. Wang et al.
also performed *in vitro* deamination assays on the B1-8 V region with recombinant AID and were
able to recapitulate the key hotspots seen *in vivo*, further arguing for the critical role of sequence
contexts in determining the mutational outcome.

2. One question arises whether the transcription of the B1-8 allele is affected and following the
removal of the two Py residues increased transcription compensates for reduction in AID intrinsic
SHM activity. Transcription levels of the various mutated versions of B1-8 alleles should be
considered.

This is an important point, and we have done as requested. As seen in new Fig. EV2, there are no
significant changes in V region nascent transcription between the different cell lines.

3. Can the authors discuss how does the AID minor groove binding property of PyPy stand with
respect to the their own findings in the discussion section. Does this mean the previous interpretation
of the AID structure was not accurate, or does it mean that only PyPy places AGCT (and not other
WRCHs) near the catalytic domain of AID. To me it wouldn't make sense that PyPy provides a proper
structural context for AGCT SHM, but not for other WRCHs.

Please note that ssDNA has no minor grooves and hence AID substrate recognition would not
involve minor groove binding.

However, this question motivated us to perform a molecular dynamics (MD) simulation of AID
binding to different ssDNA substrates used in our experiments, namely, the strong and weak AGCT
contexts in B1-8 CDR3 and FWR, respectively, and the strong CDR3 context with AGCT replaced
by two other WRCH motifs that gave reduced SHM in our assays (AACT and TGCA). The results
are in new Fig. 4. The results suggest that both the sequence context and the specific WRCH motif
modulate AID activity by altering the mode and strength of AID-ssDNA interaction in different ways.

Of note, it was previously shown using MD simulations that that a synthetic Py-only (poly-T)
sequence (but not a poly-A stretch) upstream of AGCT was able to bind the important assistant
patch of AID (PMID: 37098343). However, the endogenous CDR3 context in B1-8 lacks PyPy
motifs upstream of AGCT but can bind strongly to the assistant patch. Thus, our results highlight
that PyPy motifs are not necessary for strong binding. We thank the reviewer for raising this point
which prompted us to do the MD simulations and obtain these insightful results.

4. Authors propose that 312 AGCT SHM is a privileged position due to its distance from the TSS and
that makes it mutable, and not the presence of two Py residues. In that case, if the authors increase
the distance from TSS by 100 bps by incorporating a random sequence, it should reduce mutational
frequency at "312" AGCT? Another questions is, would the sequence leading upto the 312 AGCT
matter? This can be discussed in this paper.

We apologize for any misunderstanding. We never proposed that 311/312 AGCT is privileged due
to distance from the TSS. In the context of the B1-8 V region, this AGCT certainly appears to be
privileged, and its location is important, but as of now, we do not understand why. In other V
regions, AGCTs closer to the TSS are the most strongly mutated ones (e.g. in CDR2 in VH3-23*01
used in Wei et al., PMID: 25646473). In another study from our group, we have analyzed other V
regions (VH4-59, VH4-34 and VH3-30) and observed that the strongest mutations can be in CDR1
(VH4-59 and VH3-30) or even in FWRs (VH4-34) (doi.org/10.7554/eLife.106566). Hence, it seems
unlikely that distance from the TSS is a general contributor to the preferential mutation of some
AGCTs over others. As requested by the reviewer, we have discussed this point in the revised
manuscript.

Dear Rushad,

Thank you for the submission of your revised manuscript (EMBOJ-2025-120513R) to The EMBO Journal for our consideration, and for your patience during peer review. Your revised manuscript has been sent back to the three original referees who had previously assessed the first version of your manuscript, and we have now received their comments, which you can find below.

I am very pleased to say that, as you will see, all three referees are satisfied with the revision. They explain that their initially raised concerns have been adequately addressed in a strengthened revised manuscript, pointing out that the new MD data are very interesting and add significantly to the manuscript. All referees now recommend publication of the manuscript in The EMBO Journal.

There is only one minor remaining request by referee #1 regarding the need for better accuracy in the description of the conclusions and for acknowledging potential caveats; and a comment by referee #2 on a previously published paper and its conclusions, which could also be included in the Discussion of the manuscript. Please address these minor requests in a final version of your manuscript, and upload it to our manuscript handling system along with a point-by-point response to the referees' reports addressing these remaining comments and detailing all changes to the manuscript.

There are also a few changes and corrections from the editorial side we kindly request you to address in the final version of your manuscript, before we can move forward with its formal acceptance and publication:

- We noticed the following discrepancies between how author names have been provided in the manuscript and in the respective authors' profiles in our manuscript handling system that must be corrected: Kutti R. Vinothkumar in the manuscript vs. Vinothkumar Kutti Ragunath in the system; and Israel Tojal Da Silva in the manuscript vs. Israel Silva in the system.

- The funding information provided in the Acknowledgements section of the manuscript should be identical to that entered in our manuscript submission system; currently, the following information is missing from the online system: "Boehringer Ingelheim, The Austrian Industrial Research Promotion Agency (Headquarter Grant FFG-834223), the São Paulo Research Foundation (FAPESP), Department of Atomic Energy, Government of India, Project Identification No. RTI 4006".

- Please provide a list of up to 5 keywords (preferably broad terms to enhance the online search engine discoverability of your article) after the Abstract of your revised manuscript.

- Please move the Figure Legends to the end of the manuscript (after the list of References).

- Please make sure that all deposited data will be publicly available at the time of publication, and that the permanent, specific URLs are provided in the Data availability statement of the manuscript. The reviewer token can now be removed from this section.

- Heading "Conflicts of interest" should be renamed to "Disclosure and competing interests statement".

- The author contributions statement should be removed from the manuscript file. Instead, we use CRediT to specify the contributions of each author in the journal submission system. Please feel free to use the free text box to provide more detailed descriptions during submission. See also our guide to authors for more information:
<https://www.embopress.org/page/journal/14602075/authorguide#authorshipguidelines>.

- All main and Expanded View (EV) Figures must be uploaded individually, as separate, production-quality Figure files. Their legends must remain in the main manuscript file.

- Table EV1 and its contents (primers, oligos, and sgRNAs) should be incorporated into the single "Reagents and Tools" table that must be filled in using the template from our guide to authors (<https://www.embopress.org/page/journal/14602075/authorguide#structuredmethods>) and uploaded as a "Reagent" table.

- Please name the motif table on page 20 of your manuscript "Table 1", add a callout in the text where appropriate, and move the Table (with a brief caption) below the main Figure legends.

- Please note that EMBO press papers are accompanied online by:

- A) a short (2 sentences) summary of the findings and their significance,

- B) 2-5 short bullet points highlighting the key results, and

- C) a synopsis image in .jpg or .png format that is exactly 550 pixels wide and 300-600 pixels high (the height is variable). Please note that all text needs to be legible at the final size.

Please upload this information along with your revised manuscript (the text for A and B should be provided in a separate Word file).

- During our routine data checks, our data editors have raised the following queries regarding data, figures, and legends. Please make sure that the requests below are completely addressed in the final version of your manuscript (please highlight all changes in the revised manuscript):

1. Please provide the exact p-values in the legends of Figures 2B, 3B, 6A, C.
2. Please note that the box plots need to be defined in terms of minima, maxima, centre, bounds of box and whiskers, and percentile in the legends of Figures 1C, 2B, 3B, 6A, C; EV1 B.
3. Please note that the box plots need to be defined in terms of minima, maxima and percentile in the legend of Figure EV1 A.
4. Please note that information related to "n" is missing in the legends of Figures 1C, EV1 A, B.
5. Please note that the error bars are not defined in the legend of Figure EV2 B.

- The order of manuscript sections must be corrected as follows: Title page - Abstract and Keywords - Introduction - Results - Discussion - Methods - Data Availability - Acknowledgements - Disclosure and Competing Interests Statement - References - Figure Legends - main Tables (Table 1 in this case) - Expanded View Figure Legends.

Please also note that as part of the EMBO publications' Transparent Editorial Process, The EMBO Journal publishes online a Peer Review File along with each accepted manuscript. This File will be published in conjunction with your paper and will include the referee reports, your point-by-point response and all pertinent correspondence relating to the manuscript. You can opt out of this by letting the editorial office know (contact@embojournal.org). If you do opt out, the Peer Review File link will point to the following statement: "No Peer Review File is available with this article, as the authors have chosen not to make the review process public in this case."

We look forward to seeing a final version of your manuscript as soon as possible. Please let us know if you have any questions and use this link to submit your revision: <https://emboj.msubmit.net/cgi-bin/main.plex>.

Best regards,

Ioannis

Referee #1:

The authors have addressed my comments. I recommend publication. The findings are interesting and important also to put in context prior findings in the field.

Although not requested by me, I find very interesting the new MD data. However, the conclusions are stated a bit too strong considering that 1) the assistant patch is proposed but has not been demonstrated directly (i.e. no cocrystal with ssDNA exists for AID to be sure of the trajectory of the substrate), 2) the assistant patch was shown to be important for deamination of structured DNA, not ssDNA as modelled here, 3) the AID structures lack the C-terminal domain, an amphipathic alpha helix linked to the alpha 6 making most of the assistant patch. For the sake of accuracy I would recommend that the authors acknowledging this caveats in the discussion for the final version.

Referee #2:

The authors have addressed my specific comments in a satisfactory manner and I am satisfied that the work is rigorous and well described, with appropriate conclusions drawn. The MD simulations are provocative, add significantly to the manuscript, and help address my concern about lack of mechanistic insight. It is interesting that 5' DNA interactions with the assistance patch appear to be a key correlate of mutation efficiency. Notably, Wang et al., 2023 also found that the DNA 5' of the target C was most important for determining deamination by AID, but they attributed this to PyPy dimers, and the results of Bartl et al. here do not fit well with that. There is clearly more to learn about AID substrate selection.

Overall, the manuscript is significantly improved and represents a substantial contribution to the field that is likely to stimulate

discussion and further experiments. I support publication in EMBO J.

Referee #3:

I have no additional comments. The authors addressed my queries.

All editorial and formatting issues were resolved by the authors.

Dear Rushad,

Congratulations on an excellent manuscript! I am very pleased to inform you that it has been accepted for publication in The EMBO Journal. Thank you for comprehensively addressing the initially raised referee criticisms and the editorial requests for corrections and changes.

If you have any questions, please do not hesitate to contact the Editorial Office. Thank you for your contribution to The EMBO Journal. Working with you has been a pleasure!

Best regards,

Ioannis
